# Tailoring Self-Attention for Graph via Rooted Subtrees

**Siyuan Huang**    **Yunchong Song**    **Jiayue Zhou**    **Zhouhan Lin***
Shanghai Jiaotong University
siyuan_huang_sjtu@outlook.com    ycsong@sjtu.edu.cn    lin.zhouhan@gmail.com

## Abstract

Attention mechanisms have made significant strides in graph learning, yet they still exhibit notable limitations: local attention faces challenges in capturing long-range information due to the inherent problems of the message-passing scheme, while global attention cannot reflect the hierarchical neighborhood structure and fails to capture fine-grained local information. In this paper, we propose a novel multi-hop graph attention mechanism, named Subtree Attention (STA), to address the aforementioned issues. STA seamlessly bridges the fully-attentional structure and the rooted subtree, with theoretical proof that STA approximates the global attention under extreme settings. By allowing direct computation of attention weights among multi-hop neighbors, STA mitigates the inherent problems in existing graph attention mechanisms. Further we devise an efficient form for STA by employing kernelized softmax, which yields a linear time complexity. Our resulting GNN architecture, the STAGNN, presents a simple yet performant STA-based graph neural network leveraging a hop-aware attention strategy. Comprehensive evaluations on ten node classification datasets demonstrate that STA-based models outperform existing graph transformers and mainstream GNNs. The code is available at https://github.com/LUMIA-Group/SubTree-Attention.

## 1   Introduction

Graph Neural Networks (GNNs) have achieved remarkable performance in various tasks, such as drug discovery [35, 15], social networks [29, 31], traffic flow [11], and recommendation systems [44]. Most GNNs are based on the message-passing scheme [16], hierarchically aggregating information from multi-hop neighbors by stacking multiple layers. During this procedure, a rooted subtree can be generated for each node, representing the node's neighborhood structure. Nodes with similar neighborhood structures possess similar subtrees, which leads to similar node representations [13].

*Local attention* in graph learning can be seen as a natural combination of the message-passing scheme and the self-attention mechanism. By adaptively assigning weights among one-hop neighbors in a single layer, the local attention mechanism allows each node to focus on the most task-relevant neighbors [39, 36, 40, 3]. However, local attention limits the receptive field to one-hop neighbors. While stacking multiple local-attention layers to build a deep model can increase the receptive field, such message-passing-based deep architectures face challenges in capturing long-range dependencies [10] due to issues such as over-smoothing [5] and over-squashing [2].

On the other hand, the *global attention* mechanism originated from vanilla Transformer [38] has been widely adopted in graph learning domain [23, 12, 48, 28, 30], leveraging the fully-attentional architecture to address the aforementioned issues. However, the global attention employed by graph transformers cannot reflect the hierarchical neighborhood structure and fails to capture fine-grained local information, which is crucial in many real-world scenarios [25, 17, 46, 49]. To mitigate this deficiency, recent studies try to directly assemble global attention and message-passing-based

---

*Zhouhan Lin is the corresponding author

37th Conference on Neural Information Processing Systems (NeurIPS 2023).

models by combining GNNs and Transformers, including simultaneously applying GNNs and Transformers [50] or building Transformers on top of GNNs [33, 27, 45].

Considering the limitations of both local and global attention, we propose a multi-hop graph attention mechanism, termed SubTree Attention (STA). It allows the root node to directly attend to further neighbors in the subtree, enabling the root node to gather information from the entire rooted subtree within one layer. It provides two main advantages: (i) Compared to deep architectures with multiple local attention layers, STA avoids issues associated with the message-passing scheme such as over-smoothing and over-squashing. (ii) Compared to global attention, STA can hierarchically capture the neighborhood structure by enabling each node to focus on its own rooted subtree.

Due to the exponential growth in neighborhood size with increased hops, directly calculating attention among multi-hop neighbors becomes impractical. Meanwhile, powers of the adjacency matrix have to be stored for the calculation among multi-hop neighbors. To address these issues, we employ kernelized softmax [19] to develop an algorithm that reduces the quadratic time complexity to linear while avoiding the need to store the powers of the adjacency matrix. This efficient algorithm can be viewed as keys and values performing a random walk on the graph and eventually landing on queries.

Furthermore, we provide a theoretical analysis of STA, demonstrating that under extreme settings, STA converges to the global self-attention. As a result, STA can be regarded as a bridge between local and global attention, effectively combining the benefits of both approaches. In addition, we present the STA module with multiple attention heads. We introduce a hop-wise gating mechanism, enabling attention heads to be specialized in capturing information from specific hops.

We then propose a simple yet performant multi-hop graph attention network, named STAGNN, which can leverage multi-hop information and in the meantime acts as a fully-attentional model. As for the evaluation, we test the performance of STAGNN on ten common node classification datasets. Despite its relatively simple architecture, STAGNN consistently outperforms existing GNNs and graph transformers. Furthermore, we demonstrate that STAGNN maintains a competitive performance with an extremely deep architecture. Additional ablation studies are conducted to show the effectiveness of subtree attention even in the presence of global attention.

## 2 Background

Let $\mathcal{G} = (\mathcal{N}, \mathcal{E})$ be an undirected graph, with the associated nodes set $\mathcal{N}$ and edges set $\mathcal{E}$. We use $N = |\mathcal{N}|$ to represent the number of nodes. $\mathbf{X} \in \mathbb{R}^{N \times f}$ denotes the node feature, where $f$ denotes the number of features. Let $\mathbf{A}$ be the adjacency matrix of $\mathcal{G}$ and let $\mathbf{D}$ be the diagonal degree matrix of $\mathbf{A}$. $\mathbf{A}_{\text{sym}} = \mathbf{D}^{-1/2} \mathbf{A} \mathbf{D}^{-1/2}$ denotes the symmetric normalized adjacency matrix, while $\mathbf{A}_{\text{rw}} = \mathbf{A} \mathbf{D}^{-1}$ denotes the random walk matrix. Let $\hat{\mathbf{A}}$ be an arbitrary transition matrix, including $\mathbf{A}_{\text{rw}}$, $\mathbf{A}_{\text{sym}}$ or other matrices representing message propagation. We use $\mathbf{M}_{i:}$ and $\mathbf{M}_{:j}$ to indicate the $i^{\text{th}}$ row and the $j^{\text{th}}$ column of the matrix $\mathbf{M}$, respectively. And let $[\![0, K]\!]$ denote the set $\{0, 1, \ldots, K\}$.

### 2.1 Multi-Hop Representations

Many existing GNNs and diffusion-based models perform multi-hop message-passing in a single layer, taking advantage of multi-hop representations [8, 1, 47, 22, 51]. Among them, decoupled GCN [42] is a typical representative. One important reason for the over-smoothing problem in GCN is that neighborhood aggregation and feature transformation are coupled [4]. To address this issue, decoupled GCNs perform feature transformation and neighborhood aggregation, respectively. A general form of decoupled GCN can be described as follows [8]:

$$\mathbf{O} = \sum_{k=0}^{K} \alpha_k \text{PROPAG}_k(\mathbf{H}), \ \text{PROPAG}_k(\mathbf{H}) = \hat{\mathbf{A}}^k \mathbf{H}, \ \mathbf{H} = \text{MLP}(\mathbf{X}) \tag{1}$$

where $\{\alpha_k\}_{k \in [\![0, K]\!]}$ are the aggregation weights for different hops. $K$ represents the number of propagation steps, which also corresponds to the height of the resulting rooted subtree. Although there exist various strategies to assign weights to different hops, these methods all apply a $K$-step propagation with transition matrix $\hat{\mathbf{A}}$, which inevitably results in over-smoothing when $K$ is large.

There are relatively few attention-based methods that leverage multi-hop information. As a representative, Wang et al. (2021) [41] generalized GAT by Personalized PageRank [22]. Yet, this strategy

only employs attention scores from one-hop neighbors and necessarily acts as a low-frequency filter. In a recent work, Chen et al. (2022) [6] began with aggregating $K$-hop representations and subsequently processed them as a sequence of length $K$ using a Transformer model. Despite the decent performance, this approach still adopts a $K$-step propagation to capture multi-hop information before taking the attention mechanism into account, which leads to the aforementioned issues. This observation inspires us to incorporate the attention mechanism into the propagation phase, rather than adhering to the $K$-step propagation with transition matrix $\hat{\mathbf{A}}$.

## 2.2 Global Self-Attention and Kernelized Softmax

In graph learning domain, the Global Self-Attention function $\text{SA}(\cdot, \cdot, \cdot)$ computes a weighted sum of all positions for each node. It first projects the node feature matrix into three subspaces:

$$\mathbf{Q} = \mathbf{X}\mathbf{W}_Q, \ \mathbf{K} = \mathbf{X}\mathbf{W}_K, \ \mathbf{V} = \mathbf{X}\mathbf{W}_V \tag{2}$$

where $\mathbf{W}_Q \in \mathbb{R}^{d \times d_K}, \mathbf{W}_K \in \mathbb{R}^{d \times d_K}$ and $\mathbf{W}_V \in \mathbb{R}^{d \times d_V}$ are learnable projection matrices. Then the new representation of the $i^{\text{th}}$ node is computed as follows:

$$\text{SA}(\mathbf{Q}, \mathbf{K}, \mathbf{V})_{i:} = \frac{\sum_{j=1}^{N} \text{sim}(\mathbf{Q}_{i:}, \mathbf{K}_{j:})\mathbf{V}_{j:}}{\sum_{j=1}^{N} \text{sim}(\mathbf{Q}_{i:}, \mathbf{K}_{j:})} \tag{3}$$

where $\text{sim}(\cdot, \cdot) : \mathbb{R}^d \times \mathbb{R}^d \to \mathbb{R}$ is a function used to evaluate the similarity between queries and keys. A common form of self-attention is called softmax attention, which applies the exponential of the dot product to compute similarity: $\text{sim}(\mathbf{Q}_{i:}, \mathbf{K}_{j:}) = \exp\left(\frac{\mathbf{Q}_{i:}\mathbf{K}_{j:}^T}{\sqrt{d_K}}\right)$.

In fact, we can use an arbitrary positive-definite kernel $\kappa$ to serve as $\text{sim}(\cdot, \cdot)$. Given a selected kernel $\kappa$ and its corresponding feature map $\phi$, we can rewrite $\text{sim}(\cdot, \cdot)$ as: $\text{sim}(\mathbf{Q}_{i:}, \mathbf{K}_{j:}) = \phi(\mathbf{Q}_{i:})\phi(\mathbf{K}_{j:})^T$. Thus, Equation 3 becomes:

$$\text{SA}(\mathbf{Q}, \mathbf{K}, \mathbf{V})_{i:} = \frac{\sum_{j=1}^{N} \phi(\mathbf{Q}_{i:})\phi(\mathbf{K}_{j:})^T \mathbf{V}_{j:}}{\sum_{j=1}^{N} \phi(\mathbf{Q}_{i:})\phi(\mathbf{K}_{j:})^T} = \frac{\phi(\mathbf{Q}_{i:})\sum_{j=1}^{N} \phi(\mathbf{K}_{j:})^T \mathbf{V}_{j:}}{\phi(\mathbf{Q}_{i:})\sum_{j=1}^{N} \phi(\mathbf{K}_{j:})^T} \tag{4}$$

There are many potential choices for the feature map $\phi$. e.g., Tsai et al. (2019) [37] verified that RBF kernels perform on par with exponential kernels on neural machine translation and sequence prediction, and Choromanski et al. (2021) [9] opted for Positive Random Features (PRF).

The key advantage of Equation 4 is that all nodes share two identical summations $\sum_{j=1}^{N} \phi(\mathbf{K}_{j:})^T \mathbf{V}_{j:}$ and $\sum_{j=1}^{N} \phi(\mathbf{K}_{j:})^T$, which only need to be computed once. By doing so, we can avoid computing the full attention matrix $\{\text{sim}(\mathbf{Q}_{i:}, \mathbf{K}_{j:})\}_{i \in N, j \in N}$ and reduce the complexity from $\mathcal{O}(N^2)$ to $\mathcal{O}(N)$.

## 3 The Proposed Attention Mechanism: SubTree Attention

In this section, we present a detailed introduction to an efficient multi-hop attention mechanism called **S**ub**T**ree **A**ttention (STA). First, we give the definition of STA, followed by an efficient algorithm for computing STA based on kernelized softmax and the message-passing scheme. We then explain how multi-head STA makes attention heads hop-aware by incorporating a gate into each hop. Finally, we prove that STA approximates the global self-attention when the height of the subtree is $\mathcal{O}(\log N)$.

### 3.1 SubTree Attention

In this subsection, we give the definition of our proposed multi-hop graph attention mechanism named subtree attention. Similar to the global self-attention function $\text{SA}(\cdot, \cdot, \cdot)$, subtree attention takes queries, keys, and values as inputs and outputs new values.

We first define the method for computing attention weights among the $k^{\text{th}}$ hop neighbors, which we refer to as $\text{STA}_k$. For the $i^{\text{th}}$ node, this process can be described as follows:

$$\text{STA}_0(\mathbf{Q}, \mathbf{K}, \mathbf{V})_{i:} = \mathbf{V}_{i:},$$

$$\text{STA}_k(\mathbf{Q}, \mathbf{K}, \mathbf{V})_{i:} = \frac{\sum_{j=1}^{N} \hat{\mathbf{A}}_{ij}^k \text{sim}(\mathbf{Q}_{i:}, \mathbf{K}_{j:})\mathbf{V}_{j:}}{\sum_{j=1}^{N} \hat{\mathbf{A}}_{ij}^k \text{sim}(\mathbf{Q}_{i:}, \mathbf{K}_{j:})} \quad \forall k \in [\![1, K]\!] \tag{5}$$

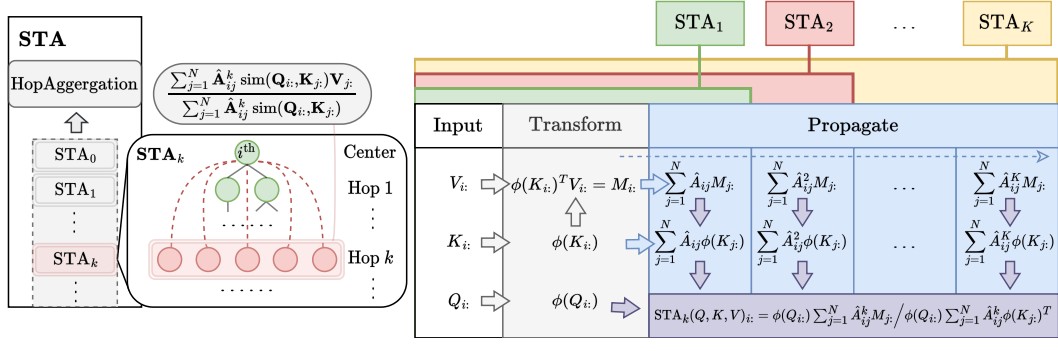

|  | (a) SubTree Attention | (b) Efficient algorithm of SubTree Attention |
|---|---|---|

Figure 1: **(Left):** Definition of SubTree Attention (STA). $\text{STA}_k$ represents that each node attends to the $k^{\text{th}}$ level of the rooted subtree, and then STA aggregates information from the entire subtree. **(Right):** An efficient algorithm for computing subtree attention. We first compute $\phi(\mathbf{K}_{i:})$ and $\phi(\mathbf{K}_{i:})^T\mathbf{V}_{i:}$ for each node, and then let $\phi(\mathbf{K}_{i:})$ and $\phi(\mathbf{K}_{i:})^T\mathbf{V}_{i:}$ perform $K$-step random walk, respectively. After each step of random walk, we compute attention weights using $\phi(\mathbf{Q}_{i:})$ and the aggregated keys and values. The computation of $\{\text{STA}_k\}_{k\in[\![1,K]\!]}$ can be seen as a nested process.

$\text{STA}_k(\mathbf{Q}, \mathbf{K}, \mathbf{V})_{i:}$ represents the $i^{\text{th}}$ node attending to its $k^{\text{th}}$ hop neighbors, which are also the $k^{\text{th}}$ level of its rooted subtree. SubTree Attention STA : $\left(\mathbb{R}^{n \times d_k}, \mathbb{R}^{n \times d_k}, \mathbb{R}^{n \times d_v}\right) \to \mathbb{R}^{n \times d_o}$ can then be calculated by aggregating the results from all levels of the rooted subtree:

$$\text{STA}(\mathbf{Q}, \mathbf{K}, \mathbf{V})_{i:} = \text{AGGR}\left(\{\text{STA}_k(\mathbf{Q}, \mathbf{K}, \mathbf{V})_{i:} \mid k \in [\![0, K]\!]\}\right) \qquad (6)$$

AGGR can be any aggregation function, such as sum, concat [18], attention-based readout [6], or the GPR-like aggregation [8] which we present in section 4.

In simple terms, $\text{STA}_k$ allows each node to attend to the $k^{\text{th}}$ level of its own subtree. Meanwhile, STA aggregates the results from all levels to gather information from the entire subtree. Figure 1 shows the process of STA.

In contrast to global attention, subtree attention enables each node to compute attention weights on its own subtree, thus taking advantage of the key insight of Message-Passing GNNs (MP-GNNs). e.g., even if the majority of nodes within the receptive fields are the same for two different nodes, they will still gather different information through subtree attention if they have different subtree structures.

## 3.2 An Efficient Algorithm for SubTree Attention

Equation 5 gives a straightforward method to calculate $\text{STA}_k$: we first compute the complete similarity matrix $\{\text{sim}(\mathbf{Q}_{i:}, \mathbf{K}_{j:})\}_{i\in N, j\in N}$ and then calculate its Hadamard product with $\hat{\mathbf{A}}^k$ to obtain $\{\hat{\mathbf{A}}^k_{ij}\text{sim}(\mathbf{Q}_{i:}, \mathbf{K}_{j:})\}_{i\in N, j\in N}$. In short, we treat $\hat{\mathbf{A}}^k$ as a mask for the similarity matrix. This algorithm exhibits two primary disadvantages: (i) The computational cost associated with calculating the entire similarity matrix is $\mathcal{O}(N^2)$. (ii) $\hat{\mathbf{A}}^k$ quickly converges to a dense matrix. Storing $\hat{\mathbf{A}}^k$ in the GPU memory for computing the Hadamard product leads to considerable space complexity. Considering these two limitations, utilizing $\hat{\mathbf{A}}^k$ as a mask for the similarity matrix is suboptimal.

We now present an efficient algorithm for subtree attention. Considering the close relationship between rooted subtrees and MP-GNNs, we leverage the message-passing scheme to implement the computation of subtree attention. By permitting keys and values to propagate along the edges, we can achieve an algorithm that has linear time complexity and avoids the need to store $\hat{\mathbf{A}}^k$.

Learning from Equation 4, we use a feature map $\phi$ to replace $\text{sim}(\cdot, \cdot)$. The choice of feature map is not the main focus of our work. Our model adopts a simple yet effective approach proposed by Katharopoulos et al. (2020) [19] that chooses $\phi(x) = elu(x) + 1$ as the feature map and demonstrates empirically that it performs on par with softmax attention. We can rewrite Equation 5 as follows:

$$\text{STA}_k(\mathbf{Q}, \mathbf{K}, \mathbf{V})_{i:} = \frac{\sum_{j=1}^N \hat{\mathbf{A}}^k_{ij}\, \phi(\mathbf{Q}_{i:})\phi(\mathbf{K}_{j:})^T\mathbf{V}_{j:}}{\sum_{j=1}^N \hat{\mathbf{A}}^k_{ij}\, \phi(\mathbf{Q}_{i:})\phi(\mathbf{K}_{j:})^T} = \frac{\phi(\mathbf{Q}_{i:}) \sum_{j=1}^N \hat{\mathbf{A}}^k_{ij}\, \phi(\mathbf{K}_{j:})^T\mathbf{V}_{j:}}{\phi(\mathbf{Q}_{i:}) \sum_{j=1}^N \hat{\mathbf{A}}^k_{ij}\, \phi(\mathbf{K}_{j:})^T} \qquad (7)$$

Note that there are two summations $\sum_{j=1}^{N} \hat{\mathbf{A}}_{ij}^{k} \, \phi(\mathbf{K}_{j:})^T \mathbf{V}_{j:}$ and $\sum_{j=1}^{N} \hat{\mathbf{A}}_{ij}^{k} \, \phi(\mathbf{K}_{j:})^T$ in Equation 7. We can think of these two summations as a kind of *message propagation*. That is to say, we first compute $\phi(\mathbf{K}_{i:})$ and $\phi(\mathbf{K}_{i:})^T \mathbf{V}_{i:}$ for each node. Then we let $\phi(\mathbf{K}_{i:})$ and $\phi(\mathbf{K}_{i:})^T \mathbf{V}_{i:}$ undergo $k$ steps message passing. Finally, we use the aggregated keys and values $\sum_{j=1}^{N} \hat{\mathbf{A}}_{ij}^{k} \, \phi(\mathbf{K}_{j:})^T \mathbf{V}_{j:}$ and $\sum_{j=1}^{N} \hat{\mathbf{A}}_{ij}^{k} \, \phi(\mathbf{K}_{j:})^T$ in conjunction with the node's own query $\phi(\mathbf{Q}_{i:})$ to complete the computation of subtree attention. Figure 1 illustrates the whole process of this efficient algorithm. When we choose $\mathbf{A}_{\mathrm{rw}}$ as the transition matrix, this process can be regarded as keys and values performing a random walk on the graph, eventually landing on different queries. Note that message passing occurs on each edge, thus reducing the computational cost from $\mathcal{O}(N^2)$ to $\mathcal{O}(|\mathcal{E}|)$. Additionally, message passing only requires the sparse adjacency matrix $\mathbf{A}$, thereby circumventing the need to store $\hat{\mathbf{A}}^k$. Furthermore, $\{\mathrm{STA}_i\}_{i \in [\![1,K]\!]}$ can be viewed as a nested process, calculated one after another.

Based on this algorithm, we can say that STA is an attempt to incorporate the message-passing scheme into the fully-attentional architecture. In fact, STA serves as a message-passing module for keys and values. In section 4, we design a novel multi-hop graph attention network employing STA for message propagation. We also provide a detailed complexity analysis in Appendix A.

### 3.3 SubTree Attention with Multiple Heads

In this subsection, we present STA with multiple attention heads. Kim et al. (2022) [20] discovered empirically that different attention heads tend to concentrate on neighbors at different hops. Certain attention heads can attend to remote nodes, while others consistently focus on nearby nodes, suggesting that attention heads can be specialized in capturing information from specific hops. To make better use of multiple attention heads in this context, we propose a hop-aware method of mixing them.

Suppose there are a total of $H$ attention heads. STA with multiple attention heads, noted as MSTA, can be described as follows:

$$\mathrm{MSTA}(\mathbf{Q}, \mathbf{K}, \mathbf{V}) = \mathrm{AGGR}\left(\{\mathrm{MSTA}_k(\mathbf{Q}, \mathbf{K}, \mathbf{V}) \mid k \in [\![0, K]\!]\}\right)$$

$$\mathrm{MSTA}_k(\mathbf{Q}, \mathbf{K}, \mathbf{V}) = \left[\mathrm{head}_k^1, \ldots, \mathrm{head}_k^H\right] \mathbf{W}_O \quad \forall k \in [\![1, K]\!], \quad \mathrm{MSTA}_0(\mathbf{Q}, \mathbf{K}, \mathbf{V}) = \mathbf{V} \quad (8)$$

$$\mathrm{head}_k^h = \hat{g}_k^h \, \mathrm{STA}_k(\mathbf{Q^h}, \mathbf{K^h}, \mathbf{V^h}) \quad \forall h \in [\![1, H]\!], \quad \hat{\boldsymbol{g}}_k = \mathrm{softmax}(\boldsymbol{g}_k)$$

where $[\,]$ denotes row-wise concatenation. $\mathbf{Q^h}, \mathbf{K^h}$, and $\mathbf{V^h}$ represent the query, key, and value matrices for the $h^{\mathrm{th}}$ head, respectively. $\mathbf{W}_O$ denotes a linear projection matrix. $\boldsymbol{g}_k \in \mathbb{R}^H$ is an $H$-dimensional vector and $g_k^h$ is its $h^{\mathrm{th}}$ element, representing the weight of the $h^{\mathrm{th}}$ attention head at the $k^{\mathrm{th}}$ hop. Compared to STA with a single attention head, we introduce in total $H \times K$ additional learnable parameters: $\{\boldsymbol{g}_i\}_{i \in [\![1,K]\!]}$. We can regard $\boldsymbol{g}_k$ as a hop-wise gate that determines the weight of each attention head at the $k^{\mathrm{th}}$ hop.

In other words, we can reconsider the multi-hop attention mechanism in terms of multi-task learning. Different attention heads are seen as different experts, while aggregating information from different hops is seen as different task. $\boldsymbol{g}_k$ signifies the process of selecting appropriate experts for each task.

### 3.4 Theoretical Analysis of SubTree Attention

MP-GNNs suffer from issues like over-smoothing or over-squashing when the height of the subtree increases. In this subsection, we theoretically demonstrate that STA avoids the issues associated with the message-passing scheme despite employing the same rooted subtree as MP-GNNs. Notice that we employ the random walk matrix $\mathbf{A}_{\mathrm{rw}}$ as the transition matrix in the STA module, *i.e.*, we have $\hat{\mathbf{A}} = \mathbf{A}_{\mathrm{rw}}$ in this subsection.

We first employ a slightly modified approach to rewrite the global self-attention module SA, which can be described as:

$$\mathrm{SA}(\mathbf{Q}, \mathbf{K}, \mathbf{V})_{i:} = \frac{\sum_{j=1}^{N} \boldsymbol{\pi}_i \, \mathrm{sim}(\mathbf{Q}_{i:}, \mathbf{K}_{j:}) \mathbf{V}_{j:}}{\sum_{j=1}^{N} \boldsymbol{\pi}_i \, \mathrm{sim}(\mathbf{Q}_{i:}, \mathbf{K}_{j:})} \quad (9)$$

where $\boldsymbol{\pi}_i = \frac{d(i)}{\sum_{j=1}^{N} d(j)}$ and $d(j)$ denotes the degree of the $j^{\mathrm{th}}$ node. Note that Equation 9 is consistent with Equation 3.

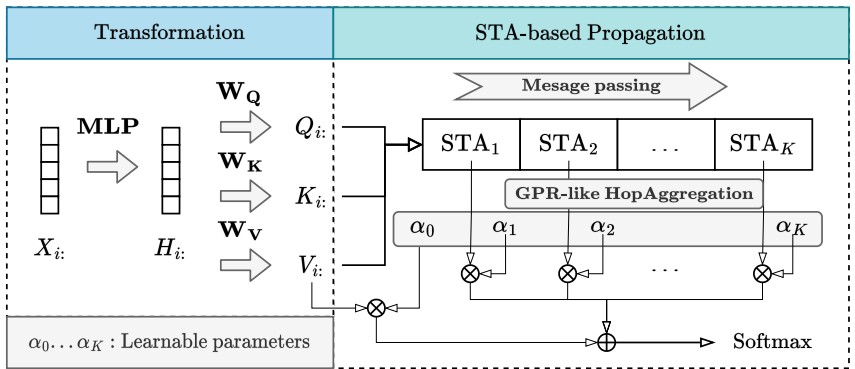

Figure 2: Overall architecture of STAGNN. STAGNN can be decomposed into two parts: Transformation and STA-based Propagation, the latter of which can be seen as an STA module using GPR-like aggregation as HopAggregation function. According to the algorithm introduced in subsection 3.2, employing STA here is equivalent to letting keys and values propagate on the graph.

Comparing Equation 5 and Equation 9, we find that the only difference between SA and $\text{STA}_k$ lies in the mask for the similarity matrix. $\text{STA}_k$ employs $\hat{\mathbf{A}}_{ij}^k$ as the mask, whereas SA employs $\boldsymbol{\pi}_i$. We now demonstrate that when the height $K$ of the rooted subtree takes the same order of magnitude as $\log(N)$, $\text{STA}_k$ can be approximately considered as SA.

**Theorem 1** *Let $\hat{\mathbf{A}} \in \mathbb{R}^{N \times N}$ denote the random walk matrix of a connected and non-bipartite graph, i.e., $\hat{\mathbf{A}} = \mathbf{A}\mathbf{D}^{-1}$. Let $1 = \lambda_1 \geq \cdots \geq \lambda_N$ be the eigenvalues of $\hat{\mathbf{A}}$. Then we have the following results:*

$$\forall i, j \in [\![1, N]\!]^2, \ \forall \epsilon > 0, \ \exists K_0 \in \mathbb{N}, \ \forall k > K_0, \ |\hat{\mathbf{A}}_{ij}^k - \boldsymbol{\pi}_i| \leq \epsilon \tag{10}$$

*And for a given $\epsilon$, the smallest $K_0$ that satisfies the condition shown in Equation 10 is at most $\mathcal{O}\left(\frac{\log \frac{N}{\epsilon}}{1-\max\{\lambda_2, |\lambda_n|\}}\right)$. If we had more information about $\mathbf{V}$, we could specify the convergence rate of $\text{STA}_k$. e.g., if $\mathbf{V}$ is computed by $\mathbf{V} = \sigma\left(\mathbf{X}\mathbf{W}_V\right)$ where $\sigma$ is a non-negative activation function, then:*

$$\forall i, j \in [\![1, N]\!]^2, \ \forall \eta \in ]0, 1[, \ \exists K_1 \in \mathbb{N}, \ \forall k > K_1, \ \frac{1-\eta}{1+\eta} \leq \frac{\text{STA}_k(\mathbf{Q}, \mathbf{K}, \mathbf{V})_{ij}}{\text{SA}(\mathbf{Q}, \mathbf{K}, \mathbf{V})_{ij}} \leq \frac{1+\eta}{1-\eta} \tag{11}$$

*holds true when none of the denominators is equal to zero. And for a given $\eta$, the smallest $K_1$ that satisfies the condition shown in Equation 11 is at most $\mathcal{O}\left(\frac{\log \frac{N}{\eta}}{1-\max\{\lambda_2, |\lambda_n|\}}\right)$.*

Equation 10 demonstrates that $\hat{\mathbf{A}}_{ij}^k$ converges to $\boldsymbol{\pi}_i$ with logarithmic complexity, indicating that under general conditions, $\text{STA}_k$ quickly tends to SA. Notice that although $\text{STA}_k$ is a multi-hop graph attention module implemented using the message-passing mechanism, it displays the characteristics of global attention when the height of the subtree is $\mathcal{O}\left(\log N\right)$. This property prevents STA from the issues associated with the message-passing scheme such as over-smoothing and over-squashing. From this perspective, subtree attention serves as a bridge connecting local and global attention. $\text{STA}_1$ plays the role of local attention, while $\text{STA}_{\mathcal{O}(\log N)}$ acts as the global self-attention. Subtree attention achieves a hierarchical attention computation by a hop-wise aggregation of $\{\text{STA}_k\}_{k \in [\![1, \mathcal{O}(\log N)]\!]}$. A detailed proof of Theorem 1 is provided in Appendix B.

## 4 The Proposed Multi-Hop Graph Attention Network: STAGNN

In this section, we present a simple yet effective multi-hop graph attention network, named STAGNN. This model is built upon decoupled GCN, but employs STA as the message-passing module instead. STAGNN can be divided into two steps: first, we apply MLP to compute queries, keys, and values for each node; then, we use STA to propagate information. Formally, it can be described as:

$$\mathbf{O} = \sum_{k=0}^{K} \alpha_k \text{STA}_k(\mathbf{Q}, \mathbf{K}, \mathbf{V}), \ \mathbf{Q} = \mathbf{H}\mathbf{W}_Q, \ \mathbf{K} = \mathbf{H}\mathbf{W}_K, \ \mathbf{V} = \mathbf{H}\mathbf{W}_V, \ \mathbf{H} = \text{MLP}(\mathbf{X}) \tag{12}$$

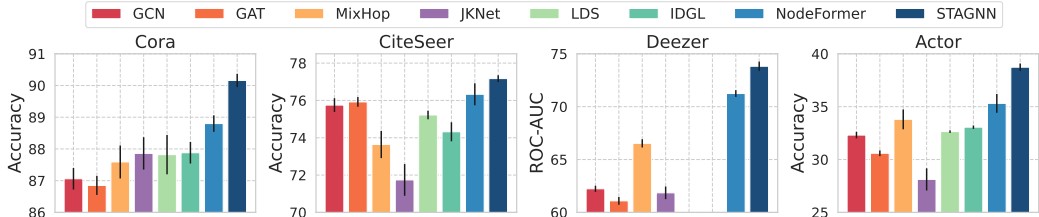

Figure 3: Comparison of four GNN baselines, three graph structure learning baselines and STAGNN on four common node classification datasets. The missing result of Deezer is due to out-of-memory.

Table 1: Comparison of four GNN baselines, five Graph Transformer baselines and STAGNN on six common node classification datasets. The best results appear in **bold**.

| Method | Pubmed | CoraFull | Computer | Photo | CS | Physics |
|---|---|---|---|---|---|---|
| GCN | $86.54_{\pm0.12}$ | $61.76_{\pm0.14}$ | $89.65_{\pm0.52}$ | $92.70_{\pm0.20}$ | $92.92_{\pm0.12}$ | $96.18_{\pm0.07}$ |
| GAT | $86.32_{\pm0.16}$ | $64.47_{\pm0.18}$ | $90.78_{\pm0.13}$ | $93.87_{\pm0.11}$ | $93.61_{\pm0.14}$ | $96.17_{\pm0.08}$ |
| APPNP | $88.43_{\pm0.15}$ | $65.16_{\pm0.28}$ | $90.18_{\pm0.17}$ | $94.32_{\pm0.14}$ | $94.49_{\pm0.07}$ | $96.54_{\pm0.07}$ |
| GPRGNN | $89.34_{\pm0.25}$ | $67.12_{\pm0.31}$ | $89.32_{\pm0.29}$ | $94.49_{\pm0.14}$ | $95.13_{\pm0.09}$ | $96.85_{\pm0.08}$ |
| GT | $88.79_{\pm0.12}$ | $61.05_{\pm0.38}$ | $91.18_{\pm0.17}$ | $94.74_{\pm0.13}$ | $94.64_{\pm0.13}$ | $97.05_{\pm0.05}$ |
| Graphormer | OOM | OOM | OOM | $92.74_{\pm0.14}$ | OOM | OOM |
| SAN | $88.22_{\pm0.15}$ | $59.01_{\pm0.34}$ | $89.83_{\pm0.16}$ | $94.86_{\pm0.10}$ | $94.51_{\pm0.15}$ | OOM |
| GraphGPS | $88.94_{\pm0.16}$ | $55.76_{\pm0.23}$ | OOM | $95.06_{\pm0.13}$ | $93.93_{\pm0.12}$ | OOM |
| NAGphormer | $89.70_{\pm0.19}$ | $71.51_{\pm0.13}$ | $91.22_{\pm0.14}$ | $95.49_{\pm0.11}$ | $95.75_{\pm0.09}$ | $\mathbf{97.34_{\pm0.03}}$ |
| STAGNN | $\mathbf{90.46_{\pm0.22}}$ | $\mathbf{72.65_{\pm0.36}}$ | $\mathbf{91.72_{\pm0.30}}$ | $\mathbf{95.64_{\pm0.27}}$ | $\mathbf{95.77_{\pm0.16}}$ | $97.09_{\pm0.18}$ |

where $\mathbf{X}$ is the input node feature and $\mathbf{O}$ is the learned representation for each node. We adopt the GPR-like aggregation [8] for HopAggregation in STA. To be precise, we assign learnable parameters $\{\alpha_k\}_{k\in[\![0,K]\!]}$ to each hop (initialized simply to 1), and then the nodes aggregate information from each hop based on these learned weights. Figure 2 shows the overall architecture of STAGNN.

Comparing Equation 1 and Equation 12, the only difference between decoupled GCN and STAGNN lies in the propagation method. Decoupled GCN relies on high powers of the normalized adjacency matrix to capture long-range information, which inevitably results in over-smoothing. In contrast, STAGNN utilizes subtree attention for message propagation, effectively learning more informative representations from multi-hop neighbors without suffering from the inherent problems associated with the message-passing scheme.

Wu et al. (2022) [43] have drawn attention to an issue named *over-normalization*: In the context of graphs with a large volume of nodes, the use of the global attention module may lead to a situation where the attention scores for the majority of nodes are nearly zero and thus resulting in gradient vanishing. Subtree attention can alleviate this problem by providing a hierarchical calculation focusing on each level of the rooted subtree instead of the whole graph.

## 5 Evaluation

We evaluate the performance of STAGNN on ten common node classification datasets, with detailed dataset information provided in Appendix C. We then verify the performance of STAGNN under extreme settings, empirically showing its capacity to tackle over-smoothing. Furthermore, We conduct an experiment that confirms the necessity of subtree attention even in the presence of global attention. Additional ablation studies are conducted for further discussions. For implementation, we fix the number of hidden channels at 64. More implementation details are presented in Appendix D. All experiments are conducted on an NVIDIA RTX4090 with 24 GB memory.

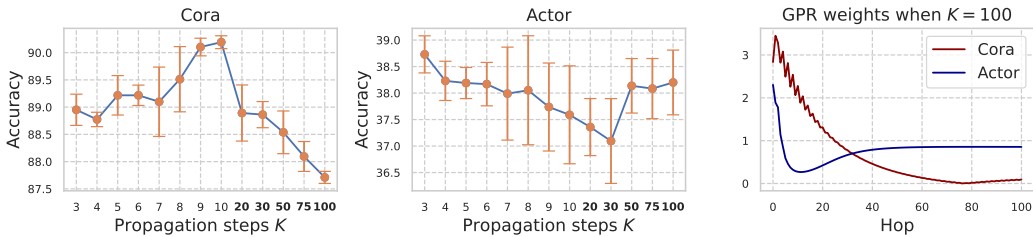

Figure 4: **(Left and Middle):** Performance of STAGNN across different heights of the rooted subtree. **(Right):** GPR weights of STAGNN when the height of the subtree $K = 100$.

.

## 5.1 Experiments on Node Classification

To compare STAGNN with a wide variety of baselines, we select two recent works [43, 6] and follow their experiment settings and their choice of baselines. The metrics of the baselines are adopted from these works [43, 6]. The code for reproduction can be found in supplementary materials.

**Comparison with Multi-Hop GNNs and Structure Learning Methods**    For a fair comparison, we strictly follow the experiment settings from Wu et al. (2022) [43]. We test the performance of STAGNN on four common datasets: Cora, Citeseer, Deezer, and Actor. The first two are homogeneous graphs, while the latter two are heterophilic [24, 52, 34]. We compare STAGNN with two mainstream GNNs: GCN [21], GAT [39], two multi-hop GNNs: JKNet [47], MixHop [1], and three graph structure learning methods: LDS [14], IDGL [7], NodeFormer [43]. We apply the same random splits with train/valid/test ratios of 50%/25%/25% as [43]. Further details can be found in Appendix D.1. Figure 3 displays the experimental results, showing that STAGNN, which features a relatively simple model architecture, outperforms all the baselines on all four datasets. This result highlights the effectiveness of STAGNN in managing both homogeneous and heterophilic graphs.

**Comparison with Decoupled GCNs and Graph Transformers**    For a fair comparison, we strictly follow the experiment settings from Chen et al. (2022) [6]. We test the performance of STAGNN on six common datasets: Pubmed, Corafull, Computer, Photo, CS and Physics. More information about these datasets can be found in Appendix C. We compare STAGNN with two mainstream GNNs: GCN [21], GAT [39], two decoupled GCNs: APPNP [22], GPRGNN [8], and five graph transformers: GT [12], Graphormer [48], SAN [23], GraphGPS [32], NAGphormer [6]. We apply the same random splits with train/valid/test ratios of 60%/20%/20% as [6]. The experimental results are shown in Table 1. STAGNN shows comparable or superior performance compared to all the baselines, which highlights the competitiveness of STAGNN when compared to existing GNNs and graph transformers.

## 5.2 Experiments on Deep STAGNN

We evaluate the performance of deep STAGNN on Cora and Actor, with the height of the subtree ranging from 3 to 100. The experimental results are presented in Figure 4. In contrast to MP-GNNs, STAGNN maintains robust performance even when the height of the subtree reaches 100. For Cora, the accuracy of STAGNN peaks at $K = 10$, demonstrating its ability to effectively collect information from a large receptive field. We further visualize the GPR weights of STAGNN when the height of the subtree is set to $K = 100$, and we observe distinct characteristics for Cora and Actor. In the case of Cora, the GPR weights exhibit a monotonic decrease, which aligns with the witnessed performance drop on Cora as the height of the subtree increases. Therefore, for Cora, we may consider keeping the height of the subtree within a reasonable range. In the case of Actor, the GPR weights eventually stabilize at a value close to 1. This finding suggests that the limiting state of $STA_k$, i.e., $SA_\pi$, is suitable for Actor, which is confirmed by robust performance of deep STAGNN on Actor. More visualizations of GPR weights can be found in Appendix E. In summary, STAGNN can achieve impressive results even with an extremely deep architecture.

Table 2: Necessity of subtree attention in the presence of global attention. We compare two scenarios: (i) Only Global attention and (ii) Global attention supplemented by subtree attention.

| Method | Pubmed | CoraFull | Computer | Photo | CS | Physics |
|---|---|---|---|---|---|---|
| GAT | $86.32_{\pm 0.16}$ | $64.47_{\pm 0.18}$ | $90.78_{\pm 0.13}$ | $93.87_{\pm 0.11}$ | $93.61_{\pm 0.14}$ | $96.17_{\pm 0.08}$ |
| Global Attn (GA) | $88.87_{\pm 0.61}$ | $62.34_{\pm 0.95}$ | $85.7_{\pm 0.52}$ | $92.92_{\pm 0.32}$ | $94.74_{\pm 0.37}$ | $96.47_{\pm 0.24}$ |
| 1-hop STA + GA | $90.16_{\pm 0.51}$ | $70.65_{\pm 0.71}$ | $91.52_{\pm 0.23}$ | $95.42_{\pm 0.47}$ | $95.49_{\pm 0.26}$ | $97.09_{\pm 0.22}$ |
| 2-hops STA + GA | $90.56_{\pm 0.49}$ | $72.24_{\pm 0.38}$ | $\mathbf{91.93_{\pm 0.35}}$ | $95.75_{\pm 0.36}$ | $95.70_{\pm 0.29}$ | $\mathbf{97.17_{\pm 0.20}}$ |
| 3-hops STA + GA | $\mathbf{90.66_{\pm 0.24}}$ | $\mathbf{72.36_{\pm 0.37}}$ | $91.89_{\pm 0.28}$ | $\mathbf{95.88_{\pm 0.31}}$ | $\mathbf{95.81_{\pm 0.15}}$ | $97.15_{\pm 0.23}$ |

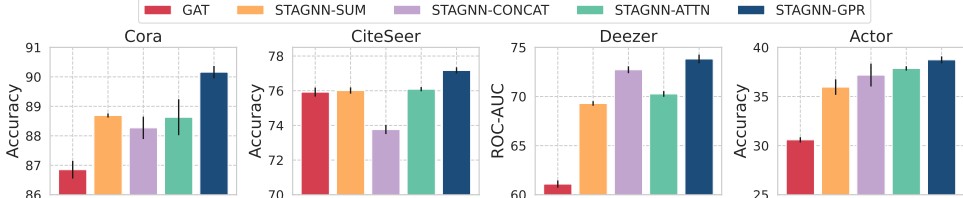

Figure 5: Comparison of different HopAggregation methods

## 5.3 Ablation Study

**Study on the Necessity of SubTree Attention in the Presence of Global Attention**   We demonstrate that $STA_k$ converges to the global self-attention, which naturally leads to a critical question:

• Is it still necessary to employ subtree attention when global attention is already present?

In this experiment, we seek to answer the question above by exploring two scenarios: (i) applying global attention independently and (ii) combining global attention with subtree attention. Formally, we extend STAGNN by replacing the STA module with global attention enhanced by 0, 1, 2, or 3 hop/hops of subtree attention. More details on the experiment settings can be found in Appendix D.2.

The experimental results are presented in Table 2. We find that incorporating subtree attention as an auxiliary to global attention significantly improves overall performance, implying that it is still necessary to employ subtree attention to capture the neighborhood structure even in the presence of global attention. This observation also inspires us that leveraging subtree attention to augment global attention could be an interesting approach for enhancing Graph Transformers.

**Study on HopAggregation methods**   In this experiment, we investigate the influence of different HopAggregation functions within STAGNN. We compare GPR-like aggregation with sum, concat [18], and attention-based readout [6]. Details of the experiment settings can be found in Appendix D.3. The experimental results are shown in Figure 5. GPR-like aggregation outperforms the alternatives on all four datasets. This observation highlights the importance of enabling nodes to adaptively learn the weight of each hop, which is the main advantage of GPR-like aggregation.

## 6   Conclusion

We propose a novel multi-hop graph attention mechanism called Subtree Attention (STA), which effectively addresses the limitations of local and global attention in graph learning. The proposed STA mechanism bridges the gap between local and global attention, hierarchically capturing neighborhood structures while addressing issues associated with the message-passing scheme. The key algorithm for computing subtree attention, utilizing kernelized softmax and the message-passing scheme, reduces the computational cost from $\mathcal{O}(N^2)$ to $\mathcal{O}(|\mathcal{E}|)$ while avoiding the need to store high powers of the adjacency matrix. This process can be approximately viewed as keys and values performing a random walk on the graph. We further prove theoretically that under extreme settings, STA approximates the global self-attention. Based on the STA module, we propose a simple yet performant multi-hop graph attention network, coined STAGNN. Comprehensive evaluations on various node classification datasets demonstrate that STAGNN outperforms mainstream GNNs and graph transformers. Ablation studies further verify the effectiveness of subtree attention, even in the presence of global attention.

**Current Limitations, Potential Impacts and Further Discussions**    In the present work, we mainly evaluate STAGNN, a novel multi-hop graph attention network incorporating STA. However, there are many other potential applications of STA, including combining STA with other graph learning methods that utilize self-attention mechanisms or supplementing global attention with subtree attention to enhance graph transformers. Furthermore, evaluating the robustness of STA and digging into its interpretability can also be part of future works. We provide a detailed discussion of potential impacts in Appendix G. And further analysis of the gate mechanism within the mixture of attention heads can be found in Appendix F.

## Acknowledgement

This work was sponsored by the National Natural Science Foundation of China (NSFC) grant (No. 62106143), and Shanghai Pujiang Program (No. 21PJ1405700).

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

# A  Complexity Analysis of STA

In this subsection, we analyze the time complexity of SubTree Attention (STA). STA has two key components: the feature map and the HopAggregation function. Both of these components offer a wide range of potential options for consideration. Different options will affect the time complexity of STA. In the following analysis, we will adopt the configuration used by STAGNN, i.e., we choose $\phi(x) = elu(x) + 1$ as the feature map and use GPR-like aggregation as the HopAggregation function.

The computation of STA can be seen as an aggregation of $\{STA_i\}_{i \in [\![1,K]\!]}$, which refer to the attention-based aggregation of each level of the rooted subtree. Therefore, we can start by analyzing the time complexity of $STA_k$.

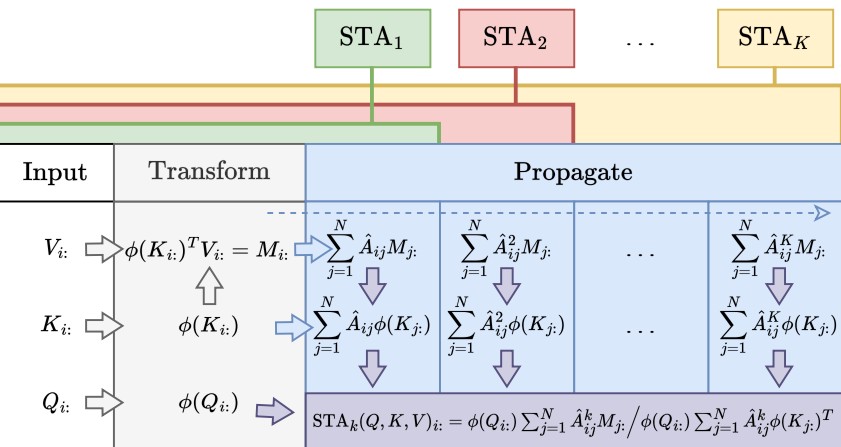

Figure 6: Efficient algorithm of SubTree Attention

The calculation of $STA_k$ can be divided into three steps. In the first step, we compute $\phi(\mathbf{K}_{i:})$ and $\phi(\mathbf{K}_{i:})^T \mathbf{V}_{i:}$ for each node. The time complexity of this step depends on the feature map. In our model, we chose $\phi(x) = elu(x) + 1$ as the feature map. Thus, the time complexity of computing $\phi(\mathbf{K}_{i:})$ is $\mathcal{O}(Nd_k)$. We also need to compute $\phi(\mathbf{K}_{i:})^T \mathbf{V}_{i:}$ for each node, the time complexity of this part is $\mathcal{O}(Nd_k d_v)$. Therefore, the overall time complexity of the first step is $\mathcal{O}(Nd_k + Nd_k d_v)$.

In the second step, we let $\phi(\mathbf{K}_{i:})$ and $\phi(\mathbf{K}_{i:})^T \mathbf{V}_{i:}$ propagate on the graph. For $STA_k$, we need to propagate $k$ times. The time complexity of propagating $\phi(\mathbf{K}_{i:})$ once is $\mathcal{O}(d_k)$, and the time complexity of $\phi(\mathbf{K}_{i:})^T \mathbf{V}_{i:}$ propagating once is $\mathcal{O}(d_k d_v)$. The message propagation occurs on each edge. Considering that there are in total $|\mathcal{E}|$ edges and $k$ times propagation, the overall time complexity of this step is $\mathcal{O}(k|\mathcal{E}|d_k + k|\mathcal{E}|d_k d_v)$.

In the third step, we use the information $\sum_{j=1}^{N} \hat{\mathbf{A}}_{ij}^k \phi(\mathbf{K}_{j:})^T \mathbf{V}_{j:}$ and $\sum_{j=1}^{N} \hat{\mathbf{A}}_{ij}^k \phi(\mathbf{K}_{j:})^T$ aggregated by each node, along with the node's own query $\phi(\mathbf{Q}_{i:})$, to complete the computation of STA. For each node, we need to calculate $\phi(\mathbf{Q}_{i:}) \sum_{j=1}^{N} \hat{\mathbf{A}}_{ij}^k \phi(\mathbf{K}_{j:})^T \mathbf{V}_{j:}$, the time complexity of this part is $\mathcal{O}(Nd_k d_v)$. At the same time, for each node, we need to calculate $\phi(\mathbf{Q}_{i:}) \sum_{j=1}^{N} \hat{\mathbf{A}}_{ij}^k \phi(\mathbf{K}_{j:})^T$, the time complexity of this part is $\mathcal{O}(Nd_k)$. So the total time complexity of this step is $\mathcal{O}(Nd_k + Nd_k d_v)$.

In summary, the total time complexity of $STA_k$ is $\mathcal{O}(2Nd_k + 2Nd_k d_v + k|\mathcal{E}|d_k + k|\mathcal{E}|d_k d_v)$.

Next, we analyze the time complexity of STA when the height of the rooted subtree is $K$. It should be noted that $\{STA_i\}_{i \in [\![1,K]\!]}$ can be viewed as a nested process, calculated one after another. Therefore, the first two steps of the above-mentioned calculation of $STA_k$ do not need to be repeated. We only need to complete the full calculation of $STA_K$ and perform the third step mentioned above $K$ times. Therefore, the time complexity of calculating STA is $\mathcal{O}((K+1)Nd_k + (K+1)Nd_k d_v + K|\mathcal{E}|d_k + K|\mathcal{E}|d_k d_v)$. In general, we can think of the time complexity of STA as $\mathcal{O}(K|\mathcal{E}|d_k d_v)$.

# B Proof for Theorem 1

## B.1 Proof for Equation 10

Let $\hat{\mathbf{A}}$ denote the random walk matrix of a connected and non-bipartite graph, and let $\mathbf{A}_{\text{sym}}$ denote the symmetric normalized adjacency matrix. Let $1 = \lambda_1 \geq \cdots \geq \lambda_N$ be the eigenvalues of $\hat{\mathbf{A}}$, which are also the eigenvalues of $\mathbf{A}_{\text{sym}}$ [26]. Let $\mathbf{v_1}, \mathbf{v_2}, \ldots, \mathbf{v_n}$ be the corresponding orthonormal eigenvectors ($\mathbf{v_1}, \mathbf{v_2}, \ldots, \mathbf{v_n}$ here are column vectors). Let $\boldsymbol{\pi}_j = \frac{d(j)}{\sum_{i=1}^{N} d(i)}$ and $d(i)$ denotes the degree of the $i^{\text{th}}$ node. $\hat{\lambda} = 1 - \max\{\lambda_2, |\lambda_n|\}$ denotes the corresponding spectral gap, and let $\mathbf{D}$ be the diagonal degree matrix. $\overrightarrow{\mathbf{1}}$ denotes an all-ones column vector.

In this subsection, we prove the following results:

$$\forall i, j \in [\![1, N]\!]^2, \ \forall \epsilon > 0, \ \exists K_0 \in \mathbb{N}, \ \forall k > K_0, \ |\hat{\mathbf{A}}_{ij}^k - \boldsymbol{\pi}_i| \leq \epsilon$$

And for a given $\epsilon$, the smallest $K_0$ that satisfies the condition above is at most $\mathcal{O}\left(\frac{\log \frac{N}{\epsilon}}{1 - \max\{\lambda_2, |\lambda_n|\}}\right)$.

We begin by considering an arbitrary distribution $\mathbf{p_i} \in \mathbb{R}^N$, which is a column vector and $\|\mathbf{p_i}\|_2 = 1$.

Notice that $\mathbf{v_1}, \mathbf{v_2}, \ldots, \mathbf{v_n}$ form an orthonormal basis, we can rewrite $\mathbf{D}^{-\frac{1}{2}}\mathbf{p_i}$ as:

$$\mathbf{D}^{-\frac{1}{2}}\mathbf{p_i} = \sum_{i=1}^{N} c_i \mathbf{v_i} \tag{13}$$

We next consider the new distribution obtained when $\mathbf{p_i}$ undergoes $k$-step random walk. Notice that $\hat{\mathbf{A}} = \mathbf{A}\mathbf{D}^{-1} = \mathbf{D}^{\frac{1}{2}}\left(\mathbf{D}^{-\frac{1}{2}}\mathbf{A}\mathbf{D}^{-\frac{1}{2}}\right)\mathbf{D}^{-\frac{1}{2}} = \mathbf{D}^{\frac{1}{2}}\mathbf{A}_{\text{sym}}\mathbf{D}^{-\frac{1}{2}}$. Thus we have:

$$
\begin{aligned}
\hat{\mathbf{A}}^k \mathbf{p_i} &= \left(\mathbf{D}^{\frac{1}{2}}\mathbf{A}_{\text{sym}}\mathbf{D}^{-\frac{1}{2}}\right)^k \mathbf{p_i} \\
&= \mathbf{D}^{\frac{1}{2}}\mathbf{A}_{\text{sym}}^k \mathbf{D}^{-\frac{1}{2}}\mathbf{p_i} \\
&= \mathbf{D}^{\frac{1}{2}}\mathbf{A}_{\text{sym}}^k \sum_{i=1}^{N} c_i \mathbf{v_i} \\
&= \sum_{i=1}^{N} c_i \mathbf{D}^{\frac{1}{2}}\mathbf{A}_{\text{sym}}^k \mathbf{v_i} \\
&= c_1 \mathbf{D}^{\frac{1}{2}}\mathbf{A}_{\text{sym}}^k \mathbf{v_1} + \sum_{i=2}^{N} c_i \mathbf{D}^{\frac{1}{2}}\mathbf{A}_{\text{sym}}^k \mathbf{v_i}
\end{aligned}
\tag{14}
$$

We now consider $c_1 \mathbf{D}^{\frac{1}{2}}\mathbf{A}_{\text{sym}}^k \mathbf{v_1}$. As we know that $\mathbf{D}^{\frac{1}{2}}\overrightarrow{\mathbf{1}}$ is an eigenvector of $\mathbf{A}_{\text{sym}}$ with eigenvalue 1. We then have:

$$\mathbf{v_1} = \frac{\mathbf{D}^{\frac{1}{2}}\overrightarrow{\mathbf{1}}}{\|\mathbf{D}^{\frac{1}{2}}\overrightarrow{\mathbf{1}}\|_2} \tag{15}$$

Notice that $\|\mathbf{D}^{\frac{1}{2}}\overrightarrow{\mathbf{1}}\|_2 = \sqrt{\sum_{i=1}^N d(i)}$. Using the fact that $\mathbf{v_1}, \mathbf{v_2}, \ldots, \mathbf{v_n}$ form an orthonormal basis and $\mathbf{D}^{-\frac{1}{2}}\mathbf{p_i} = \sum_{i=1}^N c_i \mathbf{v_i}$, we then have:

$$
\begin{aligned}
c_1 &= \left(\mathbf{D}^{-\frac{1}{2}}\mathbf{p_i}\right)^T v_1 \\
&= \mathbf{p_i}^T \mathbf{D}^{-\frac{1}{2}} \frac{\mathbf{D}^{\frac{1}{2}}\overrightarrow{\mathbf{1}}}{\|\mathbf{D}^{\frac{1}{2}}\overrightarrow{\mathbf{1}}\|_2} \\
&= \frac{\mathbf{p_i}^T \overrightarrow{\mathbf{1}}}{\|\mathbf{D}^{\frac{1}{2}}\overrightarrow{\mathbf{1}}\|_2} \\
&= \frac{1}{\|\mathbf{D}^{\frac{1}{2}}\overrightarrow{\mathbf{1}}\|_2} \\
&= \frac{1}{\sqrt{\sum_{i=1}^N d(i)}}
\end{aligned}
\tag{16}
$$

Notice that $\mathbf{A}_{\text{sym}}^k \mathbf{v_1} = \lambda_1^k \mathbf{v_1}$ and $\lambda_1 = 1$. Thus we have:

$$
\begin{aligned}
c_1 \mathbf{D}^{\frac{1}{2}} \mathbf{A}_{\text{sym}}^k \mathbf{v_1} &= c_1 \mathbf{D}^{\frac{1}{2}} \lambda_1^k \mathbf{v_1} \\
&= c_1 \mathbf{D}^{\frac{1}{2}} \mathbf{v_1} \\
&= \frac{1}{\sqrt{\sum_{i=1}^N d(i)}} \mathbf{D}^{\frac{1}{2}} \frac{\mathbf{D}^{\frac{1}{2}}\overrightarrow{\mathbf{1}}}{\|\mathbf{D}^{\frac{1}{2}}\overrightarrow{\mathbf{1}}\|_2} \\
&= \frac{\mathbf{D}\overrightarrow{\mathbf{1}}}{\sum_{i=1}^N d(i)} \\
&= \boldsymbol{\pi}
\end{aligned}
\tag{17}
$$

Considering Equation 14 and Equation 17, we have:

$$
\hat{\mathbf{A}}^k \mathbf{p_i} = \boldsymbol{\pi} + \sum_{i=2}^N c_i \mathbf{D}^{\frac{1}{2}} \mathbf{A}_{\text{sym}}^k \mathbf{v_i}
\tag{18}
$$

and immediately:

$$
\begin{aligned}
\|\hat{\mathbf{A}}^k \mathbf{p_i} - \boldsymbol{\pi}\|_2^2 &= \|\sum_{i=2}^N c_i \mathbf{D}^{\frac{1}{2}} \mathbf{A}_{\text{sym}}^k \mathbf{v_i}\|_2^2 \\
&= \|\mathbf{D}^{\frac{1}{2}} \sum_{i=2}^N c_i \mathbf{A}_{\text{sym}}^k \mathbf{v_i}\|_2^2 \\
&\leq \||\mathbf{D}^{\frac{1}{2}}\||_p^2 \|\sum_{i=2}^N c_i \mathbf{A}_{\text{sym}}^k \mathbf{v_i}\|_2^2
\end{aligned}
\tag{19}
$$

where $\||\mathbf{D}^{\frac{1}{2}}\||_p = \sup\limits_{x \in \mathbb{R}^N} \frac{\|\mathbf{D}^{\frac{1}{2}} x\|_2}{\|x\|_2} = \sqrt{d_{\text{max}}}$. Thus we have:

$$
\|\hat{\mathbf{A}}^k \mathbf{p_i} - \boldsymbol{\pi}\|_2^2 \leq d_{\text{max}} \|\sum_{i=2}^N c_i \mathbf{A}_{\text{sym}}^k \mathbf{v_i}\|_2^2
\tag{20}
$$

And using the fact that $\mathbf{v_1}, \mathbf{v_2}, \ldots, \mathbf{v_n}$ are orthonormal and $1 - \hat{\lambda} = \max\{\lambda_2, |\lambda_N|\} = \max\{|\lambda_2|, |\lambda_3|, \ldots, |\lambda_N|\}$, we then have:

$$\|\hat{\mathbf{A}}^k \mathbf{p_i} - \boldsymbol{\pi}\|_2^2 \leq d_{\max}\|\sum_{i=2}^{N} c_i \lambda_i^k \mathbf{v_i}\|_2^2$$

$$= d_{\max} \sum_{i=2}^{N} c_i^2 \lambda_i^{2k} \tag{21}$$

$$\leq d_{\max}(1 - \hat{\lambda})^{2k} \sum_{i=2}^{N} c_i^2$$

$$= d_{\max}(1 - \hat{\lambda})^{2k} \|\mathbf{D}^{-\frac{1}{2}} \mathbf{p_i}\|_2^2$$

Notice that $\|\mathbf{D}^{-\frac{1}{2}} \mathbf{p_i}\|_2^2 \leq \||\mathbf{D}^{-\frac{1}{2}}\||_p^2 \|\mathbf{p_i}\|_2^2 = \frac{1}{d_{\min}}$. Therefore:

$$\|\hat{\mathbf{A}}^k \mathbf{p_i} - \boldsymbol{\pi}\|_2^2 \leq \frac{d_{\max}}{d_{\min}}(1 - \hat{\lambda})^{2k} \tag{22}$$

and immediately:

$$\|\hat{\mathbf{A}}^k \mathbf{p_i} - \boldsymbol{\pi}\|_2 \leq \frac{d_{\max}}{d_{\min}}(1 - \hat{\lambda})^k$$

$$\leq \sqrt{N}(1 - \hat{\lambda})^k \tag{23}$$

$$\leq \sqrt{N}e^{-k\hat{\lambda}}$$

Using Cauchy–Schwarz, we then have:

$$\|\hat{\mathbf{A}}^k \mathbf{p_i} - \boldsymbol{\pi}\|_1 \leq \sqrt{N}\|\hat{\mathbf{A}}^k \mathbf{p_i} - \boldsymbol{\pi}\|_2 \leq Ne^{-k\hat{\lambda}} \tag{24}$$

In conclusion, given an arbitrarily small positive number $\epsilon$, for all $k_0$ greater than or equal to $\frac{1}{\hat{\lambda}} \log \frac{N}{\epsilon}$, the L1 norm of the difference between $\hat{\mathbf{A}}^k \mathbf{p_i}$ and the vector $\boldsymbol{\pi}$ is less than or equal to $\epsilon$. This result establishes that $\frac{1}{\hat{\lambda}} \log \frac{N}{\epsilon}$ indeed serves as an upper bound.

Notice that the vector $\mathbf{p_i}$ is an arbitrary distribution. Thus, we may consider $\mathbf{p_i}$ to be one of the $i^{\text{th}}$ unit basis vector in the $N$-dimensional space: $\{\mathbf{p_1}, \mathbf{p_2}, \ldots, \mathbf{p_N}\}$, where each vector has only one element equal to 1 (the $i^{\text{th}}$ element) and all other elements equal to 0. Thus we have $\|\hat{\mathbf{A}}^k \mathbf{p_j} - \boldsymbol{\pi}\|_1 = \sum_{i=1}^{N} |\hat{\mathbf{A}}_{ij}^k - \boldsymbol{\pi}_i|$. Then given $\epsilon > 0$, we have that:

$$\forall j \in [\![1, N]\!], \; \forall k \geq \frac{1}{\hat{\lambda}} \log \frac{N}{\epsilon}, \; \sum_{i=1}^{N} |\hat{\mathbf{A}}_{ij}^k - \boldsymbol{\pi}_i| \leq \epsilon \tag{25}$$

which demonstrates immediately the first part of Theorem 1:

$$\forall i, j \in [\![1, N]\!]^2, \; \forall k \geq \frac{1}{\hat{\lambda}} \log \frac{N}{\epsilon}, \; |\hat{\mathbf{A}}_{ij}^k - \boldsymbol{\pi}_i| \leq \epsilon \tag{26}$$

## B.2 Proof for Equation 11

In this subsection, we prove the following results: if $\mathbf{V}$ is computed by $\mathbf{V} = \sigma(\mathbf{X}\mathbf{W}_V)$ where $\sigma$ is a non-negative activation function, then:

$$\forall i, j \in [\![1, N]\!]^2, \; \forall \eta \in ]0, 1[, \; \exists K_1 \in \mathbb{N}, \; \forall k > K_1, \; \frac{1 - \eta}{1 + \eta} \leq \frac{\text{STA}_k(\mathbf{Q}, \mathbf{K}, \mathbf{V})_{ij}}{\text{SA}(\mathbf{Q}, \mathbf{K}, \mathbf{V})_{ij}} \leq \frac{1 + \eta}{1 - \eta}$$

holds true when none of the denominators is equal to zero. And for a given $\eta$, the smallest $K_1$ that satisfies the condition shown in Equation 11 is at most $\mathcal{O}\left(\frac{\log \frac{N}{\eta}}{1 - \max\{\lambda_2, |\lambda_n|\}}\right)$.

This result can, indeed, be viewed as a straightforward corollary of Equation (10). The crucial prerequisite is that all elements of the vector $\mathbf{V}$ must be positive. Importantly, there are no specific

requirements imposed on the non-negative activation function $\sigma$, meaning it can be any function that ensures non-negativity.

$\text{STA}_k$ and SA are defined as follows:

$$
\begin{aligned}
\text{STA}_k(\mathbf{Q},\mathbf{K},\mathbf{V})_{i:} &= \frac{\sum_{j=1}^{N} \hat{\mathbf{A}}_{ij}^{k}\,\text{sim}(\mathbf{Q}_{i:},\mathbf{K}_{j:})\mathbf{V}_{j:}}{\sum_{j=1}^{N} \hat{\mathbf{A}}_{ij}^{k}\,\text{sim}(\mathbf{Q}_{i:},\mathbf{K}_{j:})} \\
\text{SA}(\mathbf{Q},\mathbf{K},\mathbf{V})_{i:} &= \frac{\sum_{j=1}^{N} \boldsymbol{\pi}_i\,\text{sim}(\mathbf{Q}_{i:},\mathbf{K}_{j:})\mathbf{V}_{j:}}{\sum_{j=1}^{N} \boldsymbol{\pi}_i\,\text{sim}(\mathbf{Q}_{i:},\mathbf{K}_{j:})}
\end{aligned}
\tag{27}
$$

Equation 27 shows their form as row vectors. Their $j^{\text{th}}$ elements are:

$$
\begin{aligned}
\text{STA}_k(\mathbf{Q},\mathbf{K},\mathbf{V})_{ij} &= \frac{\sum_{t=1}^{N} \hat{\mathbf{A}}_{it}^{k}\,\text{sim}(\mathbf{Q}_{i:},\mathbf{K}_{t:})\mathbf{V}_{tj}}{\sum_{t=1}^{N} \hat{\mathbf{A}}_{it}^{k}\,\text{sim}(\mathbf{Q}_{i:},\mathbf{K}_{t:})} \\
\text{SA}(\mathbf{Q},\mathbf{K},\mathbf{V})_{ij} &= \frac{\sum_{t=1}^{N} \boldsymbol{\pi}_i\,\text{sim}(\mathbf{Q}_{i:},\mathbf{K}_{t:})\mathbf{V}_{tj}}{\sum_{t=1}^{N} \boldsymbol{\pi}_i\,\text{sim}(\mathbf{Q}_{i:},\mathbf{K}_{t:})}
\end{aligned}
\tag{28}
$$

Hence, we have:

$$
\frac{\text{STA}_k(\mathbf{Q},\mathbf{K},\mathbf{V})_{ij}}{\text{SA}(\mathbf{Q},\mathbf{K},\mathbf{V})_{ij}} = \frac{\sum_{t=1}^{N} \hat{\mathbf{A}}_{it}^{k}\,\text{sim}(\mathbf{Q}_{i:},\mathbf{K}_{t:})\mathbf{V}_{tj}}{\sum_{t=1}^{N} \boldsymbol{\pi}_i\,\text{sim}(\mathbf{Q}_{i:},\mathbf{K}_{t:})\mathbf{V}_{tj}} \times \frac{\sum_{t=1}^{N} \boldsymbol{\pi}_i\,\text{sim}(\mathbf{Q}_{i:},\mathbf{K}_{t:})}{\sum_{t=1}^{N} \hat{\mathbf{A}}_{it}^{k}\,\text{sim}(\mathbf{Q}_{i:},\mathbf{K}_{t:})}
\tag{29}
$$

For clarity, we proceed under the assumption that none of the denominators equal zero, which is reasonable considering the context. Let $\delta_{it}$ represent the difference between $\hat{\mathbf{A}}_{it}^{k}$ and $\boldsymbol{\pi}_i$: $\hat{\mathbf{A}}_{it}^{k} = \boldsymbol{\pi}_i + \delta_{it}$. Given $\eta \in\,]0,1[$, we aim to determine an upper bound of the convergence rate between $\text{STA}_k$ and SA.

Using Equation 10, we take $\epsilon = \frac{\eta}{N^2}$ and we have immediately:

$$
\forall i,t \in [\![1,N]\!]^2,\ \forall k \geq \frac{2\log\frac{N}{\eta}}{1-\hat{\lambda}},\ |\hat{\mathbf{A}}_{it}^{k} - \boldsymbol{\pi}_i| = |\delta_{it}| \leq \epsilon = \frac{\eta}{N^2}
\tag{30}
$$

We can rewrite Equation 29 as:

$$
\frac{\text{STA}_k(\mathbf{Q},\mathbf{K},\mathbf{V})_{ij}}{\text{SA}(\mathbf{Q},\mathbf{K},\mathbf{V})_{ij}} = \frac{\sum_{t=1}^{N} (\boldsymbol{\pi}_i + \delta_{it})\,\text{sim}(\mathbf{Q}_{i:},\mathbf{K}_{t:})\mathbf{V}_{tj}}{\sum_{t=1}^{N} \boldsymbol{\pi}_i\,\text{sim}(\mathbf{Q}_{i:},\mathbf{K}_{t:})\mathbf{V}_{tj}} \times \frac{\sum_{t=1}^{N} \boldsymbol{\pi}_i\,\text{sim}(\mathbf{Q}_{i:},\mathbf{K}_{t:})}{\sum_{t=1}^{N} (\boldsymbol{\pi}_i + \delta_{it})\,\text{sim}(\mathbf{Q}_{i:},\mathbf{K}_{t:})}
\tag{31}
$$

Assuming that $k \geq \frac{2\log\frac{N}{\eta}}{1-\hat{\lambda}}$. Considering the fraction $\frac{\sum_{t=1}^{N} \delta_{it}\,\text{sim}(\mathbf{Q}_{i:},\mathbf{K}_{t:})\mathbf{V}_{tj}}{\sum_{t=1}^{N} \boldsymbol{\pi}_i\,\text{sim}(\mathbf{Q}_{i:},\mathbf{K}_{t:})\mathbf{V}_{tj}}$ in the first part of Equation 31. Using Equation 30 and the fact that $\text{sim}(\mathbf{Q}_{i:},\mathbf{K}_{t:})$ and $\mathbf{V}_{tj}$ are all positive, we have:

$$
\begin{aligned}
|\sum_{t=1}^{N} \delta_{it}\,\text{sim}(\mathbf{Q}_{i:},\mathbf{K}_{t:})\mathbf{V}_{tj}| &= \sum_{t=1}^{N} |\delta_{it}|\,\text{sim}(\mathbf{Q}_{i:},\mathbf{K}_{t:})\mathbf{V}_{tj} \\
&\leq \sum_{t=1}^{N} \frac{\eta}{N^2}\,\text{sim}(\mathbf{Q}_{i:},\mathbf{K}_{t:})\mathbf{V}_{tj}
\end{aligned}
\tag{32}
$$

Notice that $\forall t \in [\![1,N]\!]$, $\boldsymbol{\pi}_i \geq \frac{1}{N^2}$. Hence, we have:

$$
|\sum_{t=1}^{N} \boldsymbol{\pi}_i\,\text{sim}(\mathbf{Q}_{i:},\mathbf{K}_{t:})\mathbf{V}_{tj}| \geq \sum_{t=1}^{N} \frac{1}{N^2}\,\text{sim}(\mathbf{Q}_{i:},\mathbf{K}_{t:})\mathbf{V}_{tj}
\tag{33}
$$

Therefore:

$$
\left| \frac{\sum_{t=1}^{N} \delta_{it}\,\text{sim}(\mathbf{Q}_{i:},\mathbf{K}_{t:})\mathbf{V}_{tj}}{\sum_{t=1}^{N} \boldsymbol{\pi}_i\,\text{sim}(\mathbf{Q}_{i:},\mathbf{K}_{t:})\mathbf{V}_{tj}} \right| = \frac{|\sum_{t=1}^{N} \delta_{it}\,\text{sim}(\mathbf{Q}_{i:},\mathbf{K}_{t:})\mathbf{V}_{tj}|}{|\sum_{t=1}^{N} \boldsymbol{\pi}_i\,\text{sim}(\mathbf{Q}_{i:},\mathbf{K}_{t:})\mathbf{V}_{tj}|} \leq \eta < 1
\tag{34}
$$

Thus we have

$$
1 - \eta \leq \frac{\sum_{t=1}^{N} (\boldsymbol{\pi}_i + \delta_{it})\,\text{sim}(\mathbf{Q}_{i:},\mathbf{K}_{t:})\mathbf{V}_{tj}}{\sum_{t=1}^{N} \boldsymbol{\pi}_i\,\text{sim}(\mathbf{Q}_{i:},\mathbf{K}_{t:})\mathbf{V}_{tj}} \leq 1 + \eta
\tag{35}
$$

Table 3: Statistics on datasets

| Dataset | Context | # Nodes | # Edges | # Features | # Classes |
|---------|---------|---------|---------|------------|-----------|
| Cora | Citation | 2,708 | 5,429 | 1,433 | 7 |
| Citeseer | Citation | 3,327 | 4,732 | 3,703 | 6 |
| Deezer | Social Connection | 28,281 | 92,752 | 31,241 | 2 |
| Actor | Co-occurrence | 7,600 | 29,926 | 931 | 5 |
| Pubmed | Citation | 19,717 | 44,324 | 500 | 3 |
| CoraFull | Citation | 19,793 | 126,842 | 8,710 | 70 |
| Computer | Co-purchasing | 13,752 | 491,722 | 767 | 10 |
| Photo | Co-purchasing | 7,650 | 238,163 | 745 | 8 |
| CS | Co-authorship | 18,333 | 163,788 | 6,805 | 15 |
| Physics | Co-authorship | 34,493 | 495,924 | 8,415 | 5 |

Considering the second part $\frac{\sum_{t=1}^{N} \boldsymbol{\pi}_i \, \mathrm{sim}(\mathbf{Q}_{i:}, \mathbf{K}_{t:})}{\sum_{t=1}^{N} (\boldsymbol{\pi}_i + \delta_{it}) \, \mathrm{sim}(\mathbf{Q}_{i:}, \mathbf{K}_{t:})}$ of Equation 31. Utilizing the same line of reasoning, we can obtain:

$$\left| \frac{\sum_{t=1}^{N} \delta_{it} \, \mathrm{sim}(\mathbf{Q}_{i:}, \mathbf{K}_{t:})}{\sum_{t=1}^{N} \boldsymbol{\pi}_i \, \mathrm{sim}(\mathbf{Q}_{i:}, \mathbf{K}_{t:})} \right| \leq \eta < 1 \tag{36}$$

and

$$\frac{1}{1+\eta} \leq \frac{\sum_{t=1}^{N} \boldsymbol{\pi}_i \, \mathrm{sim}(\mathbf{Q}_{i:}, \mathbf{K}_{t:})}{\sum_{t=1}^{N} (\boldsymbol{\pi}_i + \delta_{it}) \, \mathrm{sim}(\mathbf{Q}_{i:}, \mathbf{K}_{t:})} \leq \frac{1}{1-\eta} \tag{37}$$

Considering Equation 35, Equation 37 and Equation 31, we finally prove that:

$$\frac{1-\eta}{1+\eta} \leq \frac{\mathrm{STA}_k(\mathbf{Q}, \mathbf{K}, \mathbf{V})_{ij}}{\mathrm{SA}(\mathbf{Q}, \mathbf{K}, \mathbf{V})_{ij}} \leq \frac{1+\eta}{1-\eta} \tag{38}$$

which proves the second part of Theorem 1.

## C  Dataset Information

In this section, we present the datasets used in our experiments. These different types of data provide a robust platform to evaluate the performance of our methods.

The detailed information for each dataset is presented in Table 3. These datasets are drawn from the areas of citation networks, co-purchasing networks, co-authorship networks, and social networks: • *Citation Networks*: The citation networks datasets include Cora, Citeseer, Pubmed, and CoraFull. Nodes in these networks correspond to scientific publications, while the edges represent citations between these documents. In addition to the topological structure, each node carries a binary attribute vector, encoding the presence or absence of specific words from a pre-determined dictionary. The dimensionality of these attribute vectors varies from 1,433 in Cora to 8,710 in CoraFull. Moreover, each document node is associated with a unique class label, signifying the document's overarching scientific discipline. • *Co-authorship Networks*: We utilize the CoauthorCSDataset and Coauthor-PhysicsDataset that capture co-authorship relationships in Computer Science and Physics domains, respectively. Nodes represent individual authors and edges encode co-authorship relations, thus creating an undirected graph. • *Co-purchasing Networks*: We utilize the AmazonCoBuyComputer-Dataset and AmazonCoBuyPhotoDataset, derived from Amazon's co-purchasing network. Nodes denote products and edges symbolize frequent co-purchase incidents. Moreover, the nodes can carry diverse product-specific information. • *Social Networks*: The Deezer-Europe dataset is a dataset representing a social network of Deezer users collected via the public API in March 2020. The nodes in this network symbolize Deezer users hailing from various European countries, while the edges represent reciprocal follower relationships between these users. The features of each node are derived from the preferences of the users, specifically, the artists they have expressed an interest in. The task associated with this graph involves binary node classification, wherein the objective is to predict the user's gender. • *Co-occurrence Networks*: We utilize the Actor dataset, a type of co-occurrence network based on the Microsoft Academic Graph. Nodes represent actors, and an edge signifies their co-appearance on the same Wikipedia page.

# D Implementation Details

**Positional Encoding** We use Laplacian positional encoding to capture the structural information. As positional encoding is not the focus of our work, we use a simple approach to combine positional encoding with the original features of the nodes, which is also applied by [6]. Formally, we first calculate the eigenvectors corresponding to the smallest $m$ eigenvalues of the Laplace matrix to construct the matrix $\mathbf{P} \in \mathbb{R}^{n \times m}$. Then we take $\mathbf{X}' = [\mathbf{X}, \mathbf{P}]$ as the new input, where $[\,]$ denotes row-wise concatenation. For all the datasets, we set $m = 3$.

## D.1 Node Classification

**Training Details** We choose two recent studies [43, 6] and we adhere to their experimental configurations. The metrics for the baselines are also derived from these works [43, 6]. For Cora, Citeseer, Deezer and Actor, we apply the same random splits with train/valid/test ratios of 50%/25%/25% as [43]. We conduct 5 runs with different splits and take the mean accuracy and standard deviation for comparison. For Pubmed, Corafull, Computer, Photo, CS and Physics, we apply the same random splits with train/valid/test ratios of 60%/20%/20% as [6]. We conduct 10 runs with different splits and take the mean accuracy and standard deviation for comparison. Specifically, we utilize the ROC-AUC measure for binary classification on the Deezer dataset. For other datasets containing more than two classes, we opt for Accuracy as the metric. We employ the Adam optimizer for gradient-based optimization. The training procedure can at most repeat until a given budget of 3000 epochs and we set the patience of early stop to 200 epochs. We report the test accuracy of the epoch which has the highest accuracy on the validation set.

**Hyperparameters** For the model configuration of STAGNN, we fix the number of hidden channels at 64. We use grid search for hyper-parameter settings. The learning rate is searched within {0.001,0.01}, dropout probability searched within {0.0,0.2,0.4,0.6}, weight decay searched within {0.0001,0.0005,0.001,0.005}, height of the rooted subtree $K$ searched within {3,5,10}, number of attention heads searched within {1,2,4,6,8}. The best hyper-parameters are provided in supplementary materials.

## D.2 Study on the Necessity of SubTree Attention in the Presence of Global Attention

In this experiment, we extend STAGNN by replacing the STA module with global attention enhanced by 0, 1, 2, or 3 hop/hops subtree attention. We now present a detailed mathematical description of the experimental configurations. Formally, we compare the performance of the STAGNN-based model equipped with four different attention strategies: Global Attn Only, 1-hop STA + GA, 2-hops STA + GA and 3-hops STA + GA on six datasets: Pubmed, Corafull, Computer, Photo, CS and Physics, with the same experiment setting described in subsection D.1.

First, we calculate keys, queries and values.

$$\mathbf{Q} = \mathbf{H}\mathbf{W}_Q, \ \mathbf{K} = \mathbf{H}\mathbf{W}_K, \ \mathbf{V} = \mathbf{H}\mathbf{W}_V, \ \mathbf{H} = \mathrm{MLP}(\mathbf{X}) \tag{39}$$

Next, the output of the four different models (equipped with global attention enhanced by subtree attention of different heights) can be described as:

• *Global Attn Only*:

$$\mathbf{O} = \mathrm{SA}(\mathbf{Q}, \mathbf{K}, \mathbf{V}) \tag{40}$$

• *1-hop STA + GA*:

$$\mathbf{O} = \alpha_T \mathrm{SA}(\mathbf{Q}, \mathbf{K}, \mathbf{V}) + \sum_{k=0}^{1} \alpha_k \mathrm{STA}_k(\mathbf{Q}, \mathbf{K}, \mathbf{V}) \tag{41}$$

• *2-hops STA + GA*:

$$\mathbf{O} = \alpha_T \mathrm{SA}(\mathbf{Q}, \mathbf{K}, \mathbf{V}) + \sum_{k=0}^{2} \alpha_k \mathrm{STA}_k(\mathbf{Q}, \mathbf{K}, \mathbf{V}) \tag{42}$$

- *3-hops STA + GA*:

$$\mathbf{O} = \alpha_T \mathrm{SA}(\mathbf{Q}, \mathbf{K}, \mathbf{V}) + \sum_{k=0}^{3} \alpha_k \mathrm{STA}_k(\mathbf{Q}, \mathbf{K}, \mathbf{V}) \tag{43}$$

$\alpha_T$ here represents the coefficient of teleportation, because we can regard the global attention enhanced by subtree attention here as the random walk with teleportation. The only difference between these models is that they use subtree attention of different heights as an auxiliary to global attention. As shown in Table 2, we can observe that *2-hops STA + GA* and *3-hops STA + GA* outperform *Global Attn Only* by a large margin.

### D.3  Study on HopAggregation Methods

In this experiment, we investigate different choices of the HopAggregation functions within the STA module. We compare GPR-like aggregation with sum, concat [18], and attention-based readout [6]. We now present a detailed mathematical description of the experimental configurations. Formally, we compare the performance of the following four models: STAGNN-GPR (origin STAGNN), STAGNN-SUM, STAGNN-CONCAT and STAGNN-ATTN on four datasets: Cora, Citeseer, Deezer-Europe and Actor, with the same experiment setting described in subsection D.1.

First, we calculate keys, queries and values.

$$\mathbf{Q} = \mathbf{H}\mathbf{W}_Q, \ \mathbf{K} = \mathbf{H}\mathbf{W}_K, \ \mathbf{V} = \mathbf{H}\mathbf{W}_V, \ \mathbf{H} = \mathrm{MLP}(\mathbf{X}) \tag{44}$$

Next, the output of the four different models (STAGNN with different HopAggregation methods) can be described as:

- *STAGNN-GPR (origin STAGNN)*:

$$\mathbf{O} = \sum_{k=0}^{K} \alpha_k \mathrm{STA}_k(\mathbf{Q}, \mathbf{K}, \mathbf{V}) \tag{45}$$

- *STAGNN-SUM*:

$$\mathbf{O} = \sum_{k=0}^{K} \mathrm{STA}_k(\mathbf{Q}, \mathbf{K}, \mathbf{V}) \tag{46}$$

- *STAGNN-CONCAT*:

$$\mathbf{O} = [\mathrm{STA}_0(\mathbf{Q}, \mathbf{K}, \mathbf{V}), \mathrm{STA}_1(\mathbf{Q}, \mathbf{K}, \mathbf{V}) \ldots, \mathrm{STA}_K(\mathbf{Q}, \mathbf{K}, \mathbf{V})] \mathbf{W}_O \tag{47}$$

where $\mathbf{W}_O$ is a linear projection matrix.

- *STAGNN-ATTN*:

$$\mathbf{O} = \mathrm{STA}_0(\mathbf{Q}, \mathbf{K}, \mathbf{V}) + \sum_{k=1}^{K} \beta_k \mathrm{STA}_k(\mathbf{Q}, \mathbf{K}, \mathbf{V}),$$
$$\beta_k = \frac{\exp\left([\mathrm{STA}_0(\mathbf{Q}, \mathbf{K}, \mathbf{V}), \mathrm{STA}_k(\mathbf{Q}, \mathbf{K}, \mathbf{V})] \mathbf{W}_a^\top\right)}{\sum_{i=1}^{K} \exp\left([\mathrm{STA}_0(\mathbf{Q}, \mathbf{K}, \mathbf{V}), \mathrm{STA}_i(\mathbf{Q}, \mathbf{K}, \mathbf{V})] \mathbf{W}_a^\top\right)} \tag{48}$$

where $\mathbf{W}_a$ is a linear projection matrix and $[\ ]$ denotes row-wise concatenation.

## E  More Visualizations of GPR Weights

We conduct more visualizations of the GPR weights on Cora and Actor, with heights $K$ of the rooted subtrees ranging from 3 to 75. The results are shown in Figure 7.

In the case of Cora, we observe that as the depth $K$ of the rooted subtree increases, STA keeps increasing the GPR weights of the local neighborhood in order to preserve the local information from being covered up by the global information.

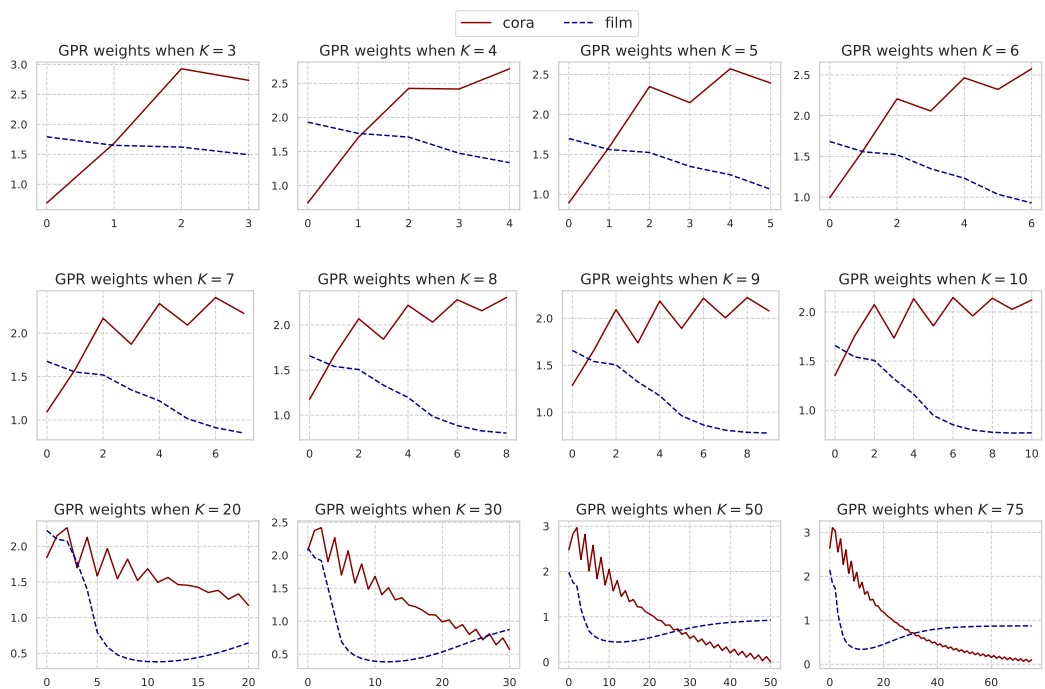

Figure 7: GPR weights of STAGNN when the heights $K$ of the subtree ranging from 3 to 75.

## F Further discussion of the Gate Mechanism within the Mixture of Attention Heads

In this subsection, we conduct an ablation study of the gate mechanism within the mixture of attention heads. The sub-tree attention module with multiple attention heads is defined as follows:

- *MSTA w/ gate vector $\boldsymbol{g}_k$, w/ softmax (origin STAGNN)*:

$$\text{MSTA}(\mathbf{Q}, \mathbf{K}, \mathbf{V}) = \text{AGGR}\left(\{\text{MSTA}_k(\mathbf{Q}, \mathbf{K}, \mathbf{V}) \mid k \in [\![0, K]\!]\}\right)$$

$$\text{MSTA}_k(\mathbf{Q}, \mathbf{K}, \mathbf{V}) = \left[\text{head}_k^1, \ldots, \text{head}_k^H\right] \mathbf{W}_O \quad \forall k \in [\![1, K]\!], \quad \text{MSTA}_0(\mathbf{Q}, \mathbf{K}, \mathbf{V}) = \mathbf{V} \quad (49)$$

$$\text{head}_k^h = \hat{g}_k^h \, \text{STA}_k(\mathbf{Q^h}, \mathbf{K^h}, \mathbf{V^h}) \quad \forall h \in [\![1, H]\!], \quad \hat{\boldsymbol{g}}_k = \text{softmax}(\boldsymbol{g}_k)$$

The hop-wise gate vector here $\boldsymbol{g}_k \in \mathbb{R}^H$ is an $H$-dimensional vector and $g_k^h$ is its $h^{\text{th}}$ element. Compared to STA with a single attention head, we introduce in total $H \times K$ additional learnable parameters: $\{\boldsymbol{g}_i\}_{i \in [\![1, K]\!]}$.

For comparison, we consider two variants.

- *MSTA w/ gate vector $\boldsymbol{g}_k$, w/o softmax*:

$$\text{MSTA}(\mathbf{Q}, \mathbf{K}, \mathbf{V}) = \text{AGGR}\left(\{\text{MSTA}_k(\mathbf{Q}, \mathbf{K}, \mathbf{V}) \mid k \in [\![0, K]\!]\}\right)$$

$$\text{MSTA}_k(\mathbf{Q}, \mathbf{K}, \mathbf{V}) = \left[\text{head}_k^1, \ldots, \text{head}_k^H\right] \mathbf{W}_O \quad \forall k \in [\![1, K]\!], \quad \text{MSTA}_0(\mathbf{Q}, \mathbf{K}, \mathbf{V}) = \mathbf{V} \quad (50)$$

$$\text{head}_k^h = g_k^h \, \text{STA}_k(\mathbf{Q^h}, \mathbf{K^h}, \mathbf{V^h}) \quad \forall h \in [\![1, H]\!]$$

- *MSTA w/o gate vector $\boldsymbol{g}_k$*:

$$\text{MSTA}(\mathbf{Q}, \mathbf{K}, \mathbf{V}) = \text{AGGR}\left(\{\text{MSTA}_k(\mathbf{Q}, \mathbf{K}, \mathbf{V}) \mid k \in [\![0, K]\!]\}\right)$$

$$\text{MSTA}_k(\mathbf{Q}, \mathbf{K}, \mathbf{V}) = \left[\text{head}_k^1, \ldots, \text{head}_k^H\right] \mathbf{W}_O \quad \forall k \in [\![1, K]\!], \quad \text{MSTA}_0(\mathbf{Q}, \mathbf{K}, \mathbf{V}) = \mathbf{V} \quad (51)$$

$$\text{head}_k^h = \text{STA}_k(\mathbf{Q^h}, \mathbf{K^h}, \mathbf{V^h}) \quad \forall h \in [\![1, H]\!]$$

The experimental results are shown in Table 4. We find that the performance of *MSTA w/ gate vector $\boldsymbol{g}_k$, w/o softmax* and *MSTA w/o gate vector $\boldsymbol{g}_k$* are almost the same, which means that using the gate

Table 4: Ablation study of the gate mechanism within the mixture of attention heads

| Method | Pubmed | CoraFull | Computer | Photo | CS | Physics |
|---|---|---|---|---|---|---|
| STAGNN (origin) | $\mathbf{90.46}_{\pm\mathbf{0.22}}$ | $\mathbf{72.65}_{\pm\mathbf{0.36}}$ | $91.72_{\pm0.30}$ | $\mathbf{95.64}_{\pm\mathbf{0.27}}$ | $\mathbf{95.77}_{\pm\mathbf{0.16}}$ | $\mathbf{97.09}_{\pm\mathbf{0.18}}$ |
| w/ gate, w/o softmax | $90.37_{\pm0.23}$ | $71.62_{\pm0.39}$ | $\mathbf{91.89}_{\pm\mathbf{0.27}}$ | $95.37_{\pm0.30}$ | $94.72_{\pm0.19}$ | $96.96_{\pm0.20}$ |
| w/o gate | $90.31_{\pm0.25}$ | $71.67_{\pm0.36}$ | $91.80_{\pm0.28}$ | $95.32_{\pm0.28}$ | $94.70_{\pm0.18}$ | $96.97_{\pm0.18}$ |

vector without softmax is approximately equivalent to not using the gate vector. In fact, on closer examination, we find that without softmax, the learned gate vector would be a vector with all equal elements, which means that it is difficult for the model to learn different weights of attention heads at each hop without the help of softmax. Additionally, we observe that for most datasets, using the gating mechanism leads to improvement of the overall performance.

# G   Potential Impacts

Besides learning better node representations, our proposed Subtree Attention (STA) has potential impacts on various aspects of graph learning. Compared to global attention, STA can help the model to better learn the hierarchical structure of the graph. Therefore, STA can be utilized as a plug-in module for designing local-aware Transformers on graph, acting as a competitor of all the GNN-assisted Transformers. STA opens new avenues for model design by combining the message-passing scheme with fully-attentional architectures, which can significantly enhance both the computational efficiency and expressive power of fully-attentional models on graph data. Furthermore, STA bridges the gap between local and global graph attention methods. This opens up possibilities for the design and application of hierarchical attention models that can leverage both local neighborhood and global structural information from graph data.

