# OpenReview forum: "Tailoring Self-Attention for Graph via Rooted Subtrees"
_NeurIPS.cc/2023/Conference — NeurIPS 2023 poster_

### Official Review · Reviewer_ds4S · 2023-07-05

**Soundness:** 3 good
**Presentation:** 2 fair
**Contribution:** 2 fair
**Rating:** 6
**Confidence:** 3

**Summary:**

The paper introduces Subtree Attention (STA), a new graph attention mechanism that overcomes limitations of existing mechanisms in graph learning. STA combines fully-attentional structure with rooted subtrees, approximating masked global attention under extreme settings. By computing attention weights among multi-hop neighbors directly, STA addresses previous mechanism problems. The authors present STAGNN, a graph neural network that utilizes STA and a hop-aware attention strategy. Extensive evaluations on ten node classification datasets show that STA-based models outperform existing graph transformers and mainstream GNNs.

**Strengths:**

1, This paper proposes a novel multi-hop graph attention mechanism, Subtree Attention (STA), which bridges the fully-attentional structure and the rooted subtree.
2, By incorporating kernelized softmax, they develop an optimized approach for STA, resulting in a linear time complexity.
3, Experiments have shown that STAGNN achieves good performance on Node classification tasks.

**Weaknesses:**

The performance improvement of NAGphormer is limited, as demonstrated in Table 1. For instance, when evaluating photos, NAGphormer achieved a score of 95.49±0.11, while STAGNN achieved 95.64±0.27. The difference between 95.49 and 95.64 is only 0.15, which is smaller than 0.27. A similar trend is observed in the case of CS. Furthermore, in Physics, STAGNN performs worse than NAGphormer. Additionally, it is worth noting that the variance of STAGNN appears to be much larger than that of NAGphormer.

**Questions:**

Why do you only evaluate STA on the node classification task? What about Graph Classification?

---

> ### Author Rebuttal · Authors · 2023-08-08
>
> Thank you for the thoughtful feedback on our manuscript. We provide the following detailed responses to your major concerns.
>
> > Q1. "The performance improvement of NAGphormer is limited, as demonstrated in Table 1. For instance, when evaluating photos, NAGphormer achieved a score of 95.49±0.11, while STAGNN achieved 95.64±0.27. The difference between 95.49 and 95.64 is only 0.15, which is smaller than 0.27. A similar trend is observed in the case of CS. Furthermore, in Physics, STAGNN performs worse than NAGphormer. Additionally, it is worth noting that the variance of STAGNN appears to be much larger than that of NAGphormer."
>
> A1. Thank you for bringing up this point.  Firstly, of the ten node classification datasets we tested, our model achieved the highest average scores on nine. This underscores the competitive performance of STAGNN.
>
> | Method     | Pubmed      | CoraFull   | Computer   | Photo      | CS         | Physics    |
> |------------|------------:|-----------:|-----------:|-----------:|-----------:|-----------:|
> | NAGphormer | 89.70±0.19  | 71.51±0.13 | 91.22±0.14 | 95.49±0.11 | 95.75±0.09 | 97.34±0.03 |
> | STAGNN     | 90.46±0.22  | 72.65±0.36 | 91.72±0.30 | 95.64±0.27 | 95.77±0.16 | 97.09±0.18 |
>
> Additionally, the performance of STAGNN and NAGphormer is tabulated above. From the table, STAGNN and NAGphormer have comparable scores on Photo, CS, and Physics, where both models already achieve very high scores (above 95%). In reality, due to inherent noise in graph datasets, pushing for even higher accuracy in such scenarios can be challenging (To the best of our knowledge, we have not observed higher scores on these three datasets in other literature). A more apt observation might be that both STAGNN and NAGphormer perform exceedingly well on these datasets.
>
> Furthermore, we would like to note that STAGNN is merely a basic application of STA. As pointed out in line 249 in our manuscript, STAGNN employs a rather simple structure. Despite this simplicity, the fact that STAGNN can surpass NAGphormer in performance further showcases the superiority of the STA mechanism.
>
> Lastly, we would like to highlight the theoretical differences between STA and NAGphormer. In brief:
> - Hop2Token initially aggregates the node representations. Subsequently, it leverages these representations to derive keys, queries, and values.
> - STA, on the contrary, computes the keys, queries, and values for each node initially and subsequently propagates the keys and values using a message-passing mechanism.
>
> The approach employed by Hop2Token, which involves message propagation on node representations, remains susceptible to challenges like over-smoothing and over-squashing. In contrast, STA's methodology of adapting graph attention through the message-passing paradigm remains immune to such issues. This ensures STA with superior theoretical properties (and, correspondingly, empirical results).
>
> We have attached a PDF in the Author Rebuttal which clearly illustrates the distinctions between STAGNN and NAGphormer. We kindly invite you to take a glance at the attached PDF.
>
> > Q2. "Why do you only evaluate STA on the node classification task? What about Graph Classification?"
>
> A2. Thanks for the reviewer's suggestion.  STAGNN is mainly designed for node classification tasks, hence we do not evaluate it on graph classification. However, STA, as a promising alternative to self-attention in the graph domain, can naturally be applied to graph classification tasks. We have conducted an additional experiment. We replace the self-attention module in the vanilla transformer with our STA module and apply this new STA+Transformer model for graph classification tasks.
>
> |Long Range Graph Benchmark (LRGB) | | Peptides-func (graph classification)| Peptides-struct (graph regression) |
> |--------------|------------:|------------:|-----------:|
> |              | #Params. |  AP ↑  |  MAE ↓ |
> | GCN|508k| 59.30±0.23| 34.96±0.13 |
> | GCNII|505k| 55.43±0.78 | 34.71±0.10 |
> | GINE |476k| 54.98±0.79| 35.47±0.45 |
> | GatedGCN |509k| 58.64±0.77|34.20±0.13 |
> | Transformer+LapPE |488k| 63.26±1.26| 25.29±0.16 |
> | **STA**+Transformer+LapPE |488k|**65.83**±**0.94**| **24.16**±**0.21**|
>
> We notice that by solely substituting the self-attention module with STA, we achieve noticeable improvements in performance. This demonstrates STA's capability in handling graph-level tasks.
>
> A more detailed analysis of this experiment can be found at the third Q&A in the Author Rebuttal.

---

### Official Review · Reviewer_K8yB · 2023-07-06

**Soundness:** 2 fair
**Presentation:** 3 good
**Contribution:** 3 good
**Rating:** 6
**Confidence:** 3

**Summary:**

This paper introduces a novel multi-hop graph attention mechanism called SubTree Attention (STA) to address the limitations of both local and global attention in Graph Neural Networks (GNNs). STA allows the root node to attend directly to further neighbors in the subtree, enabling it to gather information from the entire rooted subtree within one layer. This mechanism avoids issues associated with deep architectures using local attention layers and captures the hierarchical neighborhood structure more effectively than global attention. The authors propose an efficient algorithm based on kernelized softmax to calculate attention among multi-hop neighbors. They also provide a theoretical analysis of STA, demonstrating its convergence to a degree-masked global self-attention. The paper introduces STAGNN, a multi-hop graph attention network that integrates the STA module into decoupled GCNs, achieving superior performance compared to existing GNNs and graph transformers. Experimental evaluations on node classification datasets confirm the effectiveness of STA and the competitive performance of STAGNN.

**Strengths:**

* The authors propose a strategy to incorporate attention in multi-hop propagation GNNs and get good performances.
* Good theoretical analysis is provided.

**Weaknesses:**

* Lack of baselines (BernNet,...). The authors ignore some important baselines of propagation-based GNNs, including BernNet and successive work.
* The kernalized attention is a method from Transformer variants. Need citation.
* Datasets are not large, should include OGB-datasets, at least ogbn-arxiv

**Questions:**

* How is Graph Transformer applied to node-classification? Is it the pipeline described? Because graph for node-classification is larger than graphs for graph-level tasks? Maybe subgraph sampling is needed?
* Two questions about novelty. The STAGNN is an attention-based method to combine multi-hop information of graphs?  The difference of using vanilla transformer and STA is that STA has kernalized attention? If not, can you briefly describe the novelty here?
* Is the multi-head attention applied in STAGNN?
* If the attention will converge to degree-based global attention, why will STA work? Is the convergence achieved in training?

**Limitations:**

yes

---

> ### Author Rebuttal · Authors · 2023-08-09
>
> Many thanks for the reviewer's thoughtful feedback. We provide the following detailed responses to your major concerns.
>
> > Q1. "Lack of baselines (BernNet,..) The authors ignore some important baselines of propagation-based GNNs, including BernNet and successive work."
>
> A1. Indeed, BernNet is a compelling work in the realm of spectral GNNs. We conduct a supplementary experiment where we take BernNet as an additional baseline:
>
> Method  | Pubmed| CoraFull | Computer | Photo  | CS | Physics
> - | - | - | - | - | - | -
> BernNet | 89.12±0.33        | 67.88±0.29          | 88.8±0.34           | 94.21±0.16       | 94.64±0.10    | 96.23±0.11
> STAGNN  | 90.46±0.22        | 72.65±0.36          | 91.72±0.30          | 95.64±0.27       | 95.77±0.16    | 97.09±0.18
>
>
> We can observe that STAGNN consistently outperforms BernNet. This underscores the exceptional performance of STA.
>
> > Q2. "The kernelized attention is a method from Transformer variants. Need citation."
>
> A2. Indeed, we have cited several kernelized-related papers (line 104, 106, 149). We will take the reviewer's suggestion and offer a clearer introduction to kernelized methods.
>
> Additionally, we kindly invite the reviewer to take a glance at **the first Q&A in the Author Rebuttal** for a more detailed explanation of the role kernelized method plays in STA.
>
>
> > Q3. "Datasets are not large, should include OGB-datasets, at least ogbn-arxiv"
>
> A3. We appreciate the suggestion from the reviewer. As recommended, we conduct an additional experiment on the ogbn-arxiv dataset. We treat ogbn-arxiv as an undirected graph. The results of the experiment are as follows:
>
> Method        | ogbn-arxiv
> -|-
> MLP           | 55.50±0.23
> GCN           | 71.74±0.29
> JKNet         | 72.19±0.21
> UniMP         | 73.11±0.20
> STAGNN        | 75.42±0.35
>
>
> We can observe that even on a larger-scale dataset, STAGNN still maintains highly competitive performance.
>
> > Q4. "How is Graph Transformer applied to node classification? Is it the pipeline described? Because graph for node-classification is larger than graphs for graph-level tasks? Maybe subgraph sampling is needed?"
>
> A4. Indeed, datasets used for node classification are often larger than those for graph classification. There exist methods using graph transformers in conjunction with subgraph sampling for node classification tasks [1, 2]. In our experiments, however, we aim for a fair comparison between STA and vanilla self-attention in the context of topological data structures. Therefore, our Graph Transformer baselines do not employ these sampling techniques.
>
> While different graph transformers in our baselines have their unique designs, such as learnable positional encodings [3] and GNNs as auxiliary [4], these models employ a straightforward pipeline where each node is treated as a distinct token. They are then fed into the transformer model, yielding a final representation for each node.
>
> > Q5. "Two questions about novelty. The STAGNN is an attention-based method to combine multi-hop information of graphs? The difference of using vanilla transformer and STA is that STA has kernalized attention? If not, can you briefly describe the novelty here?"
>
> A5. This is a valuable question. We would like to highlight that STA is very different from other graph attention approaches, and **this differentiation goes beyond its use of kernelized attention**. Given that this is a critical issue raised by multiple reviewers, we have provided a detailed response in the Author Rebuttal. We kindly invite the reviewer to refer to **the first Q&A in the Author Rebuttal** for a thorough explanation.
>
> Furthermore, we kindly invite you to review the PDF attached to the Author Rebuttal above. In that pdf, we visually illustrate the novel parts of STA.
>
> > Q6. "Is the multi-head attention applied in STAGNN?"
>
> A6. Yes, in STAGNN, we employ a carefully crafted multi-head attention mechanism (section 3.3). We have also conducted an ablation study on multi-head attention within STAGNN in Appendix G.1 and demonstrated its efficacy.
>
> > Q7. "If the attention will converge to degree-based global attention, why will STA work?"
>
> A7. We fully understand the reviewer's concern. We would like to clarify that **it's not** that STA converges to degree-based global attention, but rather, $\text{STA}_k$ converges to degree-based global attention as $ k $ increases (section 3.4). And STA contains $\text{STA}_0$, $\text{STA}_1$, $\text{STA}_2$ ... And STA can finally employ various hop-aggregation strategies to adaptively adjust the weights of different hops (e.g. GPR-like aggregation used in STAGNN).
>
> We highlight that $\text{STA}_1$ functions as local attention, while $\text{STA}_k$ converges to degree-masked global attention. Thus **STA theoretically includes every intermediate step between local and global attention** (line 207).
>
> > Q8. "Is the convergence achieved in training?"
>
> A8. In fact, this convergence is intrinsic and remains unaffected by the training process. Consider both $\text{STA}_k (Q,K,V)$ and $\text{SA} (Q,K,V)$ as functions of $Q, K, V$. As illustrated in section 3.4, $\text{STA}_k (Q,K,V)$ converges to $\text{SA} (Q,K,V)$ as $k$ increases. This convergence is solely determined by $k$ and is unaffected by the values of $Q, K, V$. Conversely, the training process only modifies $Q, K, V$.
>
> Theoretically, this convergence stems from the properties of random walks on graphs. If a random walk repeats infinitely, then the probability of landing on a specific node depends on the degree of that node. This explains why  $\text{STA}_k$ converges to degree-based global attention.
>
>
> [1]. Q. Dai, et al. A Self-Attention Network based Node Embedding Model. ECML
>  2020.
>
> [2]. Z. Zhang, et al. Hierarchical Graph Transformer with Adaptive Node Sampling. NeurIPS 2022.
>
> [3]. D. Kreuzer, et al. Rethinking Graph Transformers with Spectral Attention. NeurIPS 2021.
>
> [4]. L. Rampášek, et al. Recipe for a General, Powerful, Scalable Graph Transformer. NeurIPS 2022.

---

> > ### Comment · Reviewer_K8yB · 2023-08-14
> >
> > I appreciate the authors' response. Most of my concerns are addressed. The authors need to fix their final version carefully. I will change my score to weak accept.

---

> > > ### Author Response · Authors · 2023-08-14
> > > **Many thanks**
> > >
> > > Many thanks for your positive remarks. We are committed to further refining our work and making the necessary improvements to address any concerns. Thank you for the opportunity to enhance the quality of our research.

---

### Official Review · Reviewer_smZ8 · 2023-07-06

**Soundness:** 3 good
**Presentation:** 4 excellent
**Contribution:** 3 good
**Rating:** 7
**Confidence:** 3

**Summary:**

The present manuscript proposes a novel graph attention layer, that lies in the middle of the local aggregation scheme of message passing and the global (non-structured) nature of a full-attention. The newly proposed Subtree Attention constructs for each node the similarity pairs of key and query matrices from its k-hop neighborhood.

Given the transformed matrices, they perform an efficient aggregation throughout the neighborhood hops by considering the random walk adjacency matrix as the transition matrix that is used in the graph attention. They show that given the utilization of the random walk matrix combined with a kernelized softmax function, they reduce the computational cost to $\mathcal{O}(\mathcal{E})$, where $\mathcal{E}$ the number of graph edges.

The authors prove theoretically that given a set of assumptions, the Subtree attention can approximate the global attention, showing that the proposed layer is able to tackle over-smoothing and over-squashing problems that come with MPNNs, while bridging the local with the global attention.

Their experimental study shows that STAGNN (their method) has a superior performance with respect to the state-of-the-art graph transformer models.



**Strengths:**

- The Subtree attention is a novel and interesting idea on bridging the notions of the local neighborhood attention with the global graph attention.
- The theoretical study shows that subtree attention can approximate global attention, which can be very useful for treating over-smoothing in MPNNs.
- Empirically, the model STAGNN seems to outperform its competitor baselines.
- The paper is well-written and easy to follow.

**Weaknesses:**

- The experimental study is limited to six benchmark datasets, which it is unclear whether long-range interactions play any role, or instead shallower architectures are, also, effective.
- It would be very useful testing STAGNN, also, in other datasets (e.g. from OGB), that reportedly depend on long-range interactions, or in synthetic datasets, so that the impact of the proposed model can be shown better.

**Questions:**

- Can the authors show any empirical evidence that the examined datasets can show a need for deeper models? In other words, do these datasets consist of any long-range interactions, or do they consist only of shallow interactions?

**Limitations:**

The authors make a discussion on the supplementary material over possible extensions and ways to improve the STAGNN by incorporating it to more models.

---

> ### Author Rebuttal · Authors · 2023-08-08
>
> We sincerely appreciate the reviewer's constructive feedback and positive remarks on our work. We provide the following detailed responses to your major concerns.
>
> In order to organize our response logically, we jointly address the first weakness and the question raised by the reviewer.
>
> >Q1. "The experimental study is limited to six benchmark datasets, which it is unclear whether long-range interactions play any role, or instead shallower architectures are, also, effective."
>
> > "Can the authors show any empirical evidence that the examined datasets can show a need for deeper models? In other words, do these datasets consist of any long-range interactions, or do they consist only of shallow interactions?"
>
> A1. In fact, we have conducted evaluations on 10 datasets in our paper. Kindly refer to Figure 3 (4 datasets) and Table 1 (6 datasets). These datasets come from real-world scenarios. Given the intricate relationships in real-world networks, it appears challenging to definitively state whether they rely more on long-range or short-range relationships or a blend of both. However, we would like to note that these datasets are frequently used as standards in assessing the capabilities of GNNs. In fact, numerous graph neural network methodologies [1,2,3] have utilized these datasets (or part of them) as performance benchmarks. Testing on these datasets offers a fair representation of our model's performance.
>
> At the same time, we do have a piece of indirect evidence that reflects the level of involvement of the long-range interaction: **the GPR weights** within STAGNN. GPR weights are the weights learned for each hop when STAGNN adaptively aggregates the information across different hops. They can be considered as a series of 'importance scores' that the model deduces during training for different hops.
>
> Let us give an example. The GPR weights for the Cora and Actor datasets are visualized in Figure 7 (Appendix section F). For instance, in the Cora dataset, we observe an increasing trend in the GPR weights over hops ranging from 0 to 10. This suggests that the model values relationships that span longer distances than relationships of just one or two hops. In contrast, for the Actor dataset, there's a noticeable decline in GPR weights over hops ranging from 0 to 10, highlighting the model's preference for shorter relationships. In summary, the GPR weights adaptively learned by STAGNN during the training process could serve as indirect evidence of which order of neighbors (longer or shorter relationships) the model primarily focuses on. We believe this also showcases the strong interpretability of STA.
>
> We appreciate the valuable question from the reviewer. While these datasets might not clearly show STAGNN's ability to handle long-range interactions, we've added an additional experiment using the Long Range Graph Benchmark dataset (see our answer to question 2).
>
> Finally, we would like to note that STA isn't born for handling long-range interactions. STA is born to be an all-around player. We theoretically prove that STA includes both local and global attention (approximately), as well as every intermediate step between local and global attention (line 207), and thus STA is more like an all-around hand for both long and short interactions.
>
> >Q2. It would be very useful testing STAGNN, also, in other datasets (e.g. from OGB), that reportedly depend on long-range interactions, or in synthetic datasets, so that the impact of the proposed model can be shown better.
>
> A2. We greatly appreciate the suggestion provided by the reviewer. Based on this recommendation, we opt for two graph datasets from the **Long Range Graph Benchmark (LRGB)** to conduct an experiment [4].  LRGB includes 5 graph learning datasets that require long-range interactions to achieve strong performance in a given task.
>
> We aim to demonstrate that STA (rather than STAGNN) possesses the capability to capture long-range interactions. Thus, we replaced the self-attention module in the vanilla transformer with our STA module and employed this new STA+Transformer model for long-range interaction learning tasks.
>
> We elaborate on this experiment in detail in the Author Rebuttal. We kindly invite you to take a look at the **third Q&A in the Author Rebuttal**. Below are the experimental results:
>
> |Long Range Graph Benchmark (LRGB) | | Peptides-func (graph classification)| Peptides-struct (graph regression) |
> |--------------|------------:|------------:|-----------:|
> |              | #Params. |  AP ↑  |  MAE ↓ |
> | GCN|508k| 59.30±0.23| 34.96±0.13 |
> | GCNII|505k| 55.43±0.78 | 34.71±0.10 |
> | GINE |476k| 54.98±0.79| 35.47±0.45 |
> | GatedGCN |509k| 58.64±0.77|34.20±0.13 |
> | Transformer+LapPE |488k| 63.26±1.26| 25.29±0.16 |
> | **STA**+Transformer+LapPE |488k|**65.83**±**0.94**| **24.16**±**0.21**|
>
> By solely substituting the self-attention module with STA, we achieved noticeable improvements in performance. This highlights STA's ability to handle long-range interactions effectively.
>
>
> [1]. E. Chien, et al. Adaptive Universal Generalized PageRank Graph Neural Network. ICLR 2021.
>
> [2]. Q. Wu, et al. NodeFormer: A Scalable Graph Structure Learning Transformer for Node Classification. NeurIPS 2022.
>
> [3]. J. Chen, et al. NAGphormer: A Tokenized Graph Transformer for Node Classification in Large Graphs. ICLR 2023.
>
> [4]. VP. Dwivedi, et al. Long Range Graph Benchmark. NeurIPS 2022 Track on Datasets and Benchmarks.

---

### Official Review · Reviewer_sk7A · 2023-07-06

**Soundness:** 3 good
**Presentation:** 2 fair
**Contribution:** 2 fair
**Rating:** 5
**Confidence:** 3

**Summary:**

In this paper, the authors propose a novel Graph Transformer called STA-GNN (SubTree-attention-GNN) to address the over-smoothing and over-squashing in the message-passing scheme. Different from the previous Graph Transformer, STA-GNN's attention mechanism is called SubTree Attention (STA), which computes a node's attention according to different levels of its subtree. To overcome the exponentially growing computation cost STA brings, the authors leverage the linearized attention technique raised by [19]. Experiments show that STAGNN outperforms SOTA GNN on node classification tasks.

**Strengths:**

1. The authors provide a thorough demonstration of STA's workflow with a theoretical analysis of how STA addresses over-smoothing and over-squishing. Besides. The authors propose an effective computing algorithm, which decreases STA time complexity.

2. The experiment result outperforms SOTA GCN and Graph Transformer.

3. Ablation studies show that STA is beneficial to global attention in graph learning.

**Weaknesses:**

1. STA's timing performance is unclear. The authors leverage kernelized Softmax to design an efficient algorithm with time complexity O(epsilon) (the number of edges). However, it is hard to compare STA with Graph transformers' (O(N) where N is the number of nodes) from the time complexity aspect, since the number of nodes and edges are independent in an undirected graph. This paper lacks experiments on STA's and Graph Transformers' timing performance. Without evidence, people will worry about if STA can achieve superior performance within a reasonable time limit.

2. Analysis of STA's space complexity is missing. STA efficient algorithm reduces the number of calculations by storing and re-using previous calculation results for future calculation. But there is no discussion on the potential problem of STA's memory cost. A memory cost comparison can better show the efficiency of STA.

3. Hop2Token used in NAGphormer [1] also learns the graph at the level of hops like STA. The experiment results against with NAGphormer does not seem very superior, especially when the experiment is not comprehensive to include a comparison of time and memory cost.

4. Equation (5) doesn't match the context in Figure 1 (a). N is defined as the number of nodes at the start of Section 2. However, figure 1 (a) says the STA_{k} should only attend the k-th hop neighbors, which is conflict with Equation (5).

5. Lack of explanation when introducing the rewriting from equation (3) to equation (4) (in Sec. 2.2). It is not obvious how to rewrite a similarity function with a selected kernel and feature map. [2] is about how to rewrite the similarity function using a linear kernel, which should be cited here.

[1] J. Chen, K. Gao, G. Li, and K. He. Nagphormer: Neighborhood aggregation graph transformer for node classification in large graphs. CoRR, abs/2206.04910, 2022.

[2] A. Katharopoulos, A. Vyas, N. Pappas, and F. Fleuret. Transformers are rnns: Fast autoregressive
transformers with linear attention. In Proceedings of the 37th International Conference on Machine Learning, ICML 2020, 13-18 July 2020, Virtual Event, volume 119 of Proceedings of Machine Learning Research, pages 5156–5165. PMLR, 2020.

**Questions:**

1. In section 5.2, the authors claim "In contrast to MP-GNNs, STAGNN maintains robust performance even when the height of the subtree reaches 100". Where is the source of "the performance of MP-GNNs doesn't maintain robust when the height of the subtree reaches 100"?

**Limitations:**

No limitations are mentioned in this paper.

---

> ### Author Rebuttal · Authors · 2023-08-08
>
> Thank you for the detailed comments and valuable questions. We provide details to clarify your major concerns.
>
> > Q1 & Q2. (Concerns related to time and space complexity)
>
> A1 & A2.
> Firstly, we would like to highlight one point: while we indeed introduce an efficient algorithm, the primary motivation of this algorithm is to **address the implementation challenges associated with STA** (line 136). The goal **is not** to make STA faster or lighter compared to other graph attention mechanisms.
>
> STA is specifically tailored for graph data with topological structures, which results in a more intricate computation process. We craft the algorithm to ensure the feasibility of STA,  not with the primary intent of outperforming other methods in computational efficiency. Thus, we do not compare time and space complexity with other graph learning methods in our paper.
>
> We appreciate the reviewer for pointing out the importance of this aspect. Heeding the advice of the reviewer, we conduct an additional experiment focusing on time and space complexity:
>
> |              | Cora|        | Actor|        | Deezer|        |
> |--------------|:------------|:-------|:-----------|:-------|:-----------|:-------|
> |              | Memory(MB)  | Time(s/epoch)| Memory(MB) | Time(s/epoch)|Memory(MB) | Time(s/epoch)|
> |GCN|1,058| 0.007| 1,136 | 0.018| 4,012| 0.103|
> |GraphGPS|3,280| 0.043|  4,583 | 0.075|OOM| OOM|
> |STAGNN|1,263| 0.008|  1,341 | 0.025| 4,896| 0.136|
>
> It's worth noting that the training time and memory required by STAGNN are similar to GCN. **This is not a coincidence.** This is due to our proposed efficient algorithm for STA (section 3.2) where we leverage **the Message-passing scheme** to compute graph attention, which ensures that **STAGNN has similar time and space complexity compared to vanilla Message-passing GNNs (especially, GCN)**, which is completely acceptable.
>
> Additionally, we'd like to note that there is potential for making STA **even faster**. Since STA employs a Message-passing scheme to compute attention, it can be combined with existing GNN acceleration techniques (e.g. [1]) to architect an even quicker graph attention methodology. We are willing to delve deeper into it in future work.
>
> > Q3. "Hop2Token used in NAGphormer also learns the graph at the level of hops like STA. The experiment results against with NAGphormer does not seem very superior, especially when the experiment is not comprehensive to include a comparison of time and memory cost."
>
> A3. Since this is a common question raised by multiple reviewers, we provide a unified reply in the Author Rebuttal at the top of this page. We kindly invite you to take a glance at **the second Q&A of the Author Rebuttal**.
>
> Furthermore, we kindly invite you to take a look at the pdf attached in the Author Rebuttal. This pdf clearly illustrates the superior theoretical properties of STA compared to NAGphormer.
>
> > Q4. "Equation (5) doesn't match the context in Figure 1 (a). N is defined as the number of nodes at the start of section 2. However, figure 1 (a) says the $\text{STA}_k$ should only attend the $k$-th hop neighbors, which conflicts with Equation (5).
>
> A4. Equation (5) **does match** Figure 1 (a). In fact, Equation (5) can be found at the top-right corner of Figure 1 (a).
>
> We understand the reviewer's concern regarding the definition of "N". Please notice that within Equation (5), $A^k$ is incorporated, which implies $\text{STA}_k$ should only attend to the $k$-th hop neighbors".
>
> We hope that this explanation has addressed the reviewer's concern.
>
> > Q5. "Lack of explanation when introducing the rewriting from equation (3) to equation (4) (in section 2.2). It is not obvious how to rewrite a similarity function with a selected kernel and feature map. [2] is about how to rewrite the similarity function using a linear kernel, which should be cited here."
>
> A5. In fact, [2] has been cited in our paper (line 149). Additionally, we have cited multiple kernelized-related works (line 104,106). We will take the reviewer's suggestion and cite these papers at the beginning of our introduction to kernel methods.
>
> > Q6. "In section 5.2, the authors claim "In contrast to MP-GNNs, STAGNN maintains robust performance even when the height of the subtree reaches 100". Where is the source of "the performance of MP-GNNs doesn't maintain robust when the height of the subtree reaches 100"?"
>
> A6. Our claim about the depth limitation of MP-GNNs stems from the widely accepted consensus within the graph neural network community. In [3], it's mentioned that "GNNs suffer a model depth limitation—they tend to perform increasingly worse on classifying graph nodes as the model gets deeper." Another work [4] highlights that "GNNs are susceptible to a bottleneck when aggregating messages across a long path." (Here, "long path" is synonymous with deep subtree) From a more theoretical perspective, the study [5] mentions in Theorem 2 that there's "an exponential information loss of GCNs in terms of the layer size."
>
> In fact, there's no need for such a large number as 100. 7 is enough. [6] has empirically observed that GCN, when evaluated on the Cora dataset, exhibits an accuracy of approximately 37% upon reaching a subtree depth of 7. In contrast, our STAGNN showcases a performance of 89.2% at the same subtree depth.
>
>
> [1]. H. Zeng, et al. Graphsaint: Graph sampling based inductive learning method. ICLR 2020.
>
> [2]. A. Katharopoulos, et al. Transformers are rnns: Fast autoregressive transformers with linear attention. PMLR, 2020.
>
> [3]. K. Zhou, et al. Understanding and resolving performance degradation in deep graph convolutional networks. CIKM 2021.
>
> [4]. U. Alon, et al. On the Bottleneck of Graph Neural Networks and its Practical Implications. ICLR 2021.
>
> [5]. K. Oono, et al. Graph Neural Networks Exponentially Lose Expressive Power for Node Classification. ICLR 2020.
>
> [6]. M. Liu, et al. Towards deeper graph neural networks. KDD 2020.

---

> > ### Comment · Reviewer_sk7A · 2023-08-19
> >
> > Thanks to the authors for the response to my concerns. My major concerns are well addressed. I'm willing to increase my score to 5.

---

> > > ### Author Response · Authors · 2023-08-20
> > > **Thank you for your positive feedback**
> > >
> > > We appreciate the reviewer's positive feedback. We remain dedicated to ongoing improvement. And we thank the reviewer for helping us improve the quality of our paper.

---

### Author Rebuttal · Authors · 2023-08-08

We extend our sincere gratitude to the reviewers for their invaluable feedback on our work.

We are delighted to see comments such as "**The Subtree attention is a novel and interesting idea**" (Reviewer smZ8), "**Good theoretical analysis is provided.**" (Reviewer K8yB), and "**can be very useful for treating over-smoothing in MPNNs**" (Reviewer smZ8).

We have carefully considered each comment. Below, we address three common questions raised by some reviewers:

> Q1. How does STA differ from the vanilla graph transformer and NAGphormer? Where is its novelty?

A1. The novelty of STA proposed in this paper can be summarized in two main points.

Firstly, STA presents a novel method for graph attention calculation.  We demonstrate that **STA theoretically includes both local and global attention, as well as every intermediate step between local and global attention** (line 207).  $\text{STA}_1$ functions as local attention, while $\text{STA}_k$ converges to degree-based global attention as $ k $ increases (section 3.4). And STA contains $\text{STA}_0$, $\text{STA}_1$ ...  $\text{STA}_k$ and thus can better capture the hierarchical neighborhood structure than existing graph attention methods.

We provide an intuitive comparison of STA, vanilla graph transformer, and NAGphormer in the attached PDF (You can view it at the bottom of this Author Rebuttal). **We kindly invite you to take a minute to review the attached PDF**.

Secondly, while the idea behind STA is straightforward, its computation is more intricate compared to other vanilla graph attention. We innovatively use kernelized methods to address this issue. To be clear, we **are not** the pioneers in applying kernelized methods in the graph attention domain, but **we are the first to fuse message-passing frameworks with graph attention computations using kernelized methods**. In simpler terms, we initially compute the key, value, and query for each node, followed by the propagation of the key and value across the graph. This idea of computing graph attention is novel and, in our belief, holds vast potential for extensions, such as integration with edge sampling methods.

> Q2. STAGNN doesn't seem to have achieved significant improvements over NAGphormer?

A2. Firstly, of the ten node classification datasets we tested, our model achieved the highest average scores on nine. This underscores the competitive performance of STAGNN.
| Method     | Pubmed      | CoraFull   | Computer   | Photo      | CS         | Physics    |
|------------|------------:|-----------:|-----------:|-----------:|-----------:|-----------:|
| NAGphormer | 89.70±0.19  | 71.51±0.13 | 91.22±0.14 | 95.49±0.11 | 95.75±0.09 | 97.34±0.03 |
| STAGNN     | 90.46±0.22  | 72.65±0.36 | 91.72±0.30 | 95.64±0.27 | 95.77±0.16 | 97.09±0.18 |

Additionally, the performance of STAGNN and NAGphormer is tabulated above. From the table, STAGNN and NAGphormer have comparable scores on Photo, CS, and Physics, where both models already achieve very high scores (above 95%). In reality, due to inherent noise in graph datasets, pushing for even higher accuracy in such scenarios can be challenging (To the best of our knowledge, we have not observed higher scores on these three datasets in other literature). A more apt observation might be that both STAGNN and NAGphormer perform exceedingly well on these datasets.

Lastly, we would like to note that STAGNN is merely a basic application of STA. As pointed out in line 249 in our manuscript, STAGNN employs a rather simple structure. Despite this simplicity, the fact that STAGNN can surpass NAGphormer in performance further showcases the superiority of the STA mechanism.

> Q3. The main contribution of this paper is to accelerate Graph Attention with kernelized methods?

A3. The answer is **NO**. We emphasize that our primary contribution is **SubTree Attention**. The motivation behind STA is its superior capability and robust theoretical properties. Beyond this main objective, we employ a kernelized approach to devise a linear complexity algorithm for computing STA and introduce STAGNN. Compared to STA, both the efficient computation of STA and STAGNN are secondary contributions.

STA can do more. STA can be employed to enhance existing attention-based graph methods. This potential is also echoed by "Ablation studies show that STA is beneficial to global attention in graph learning." (Reviewer sk7A).

To better showcase the potential of STA, we conduct an additional experiment. We replace the self-attention module in the vanilla transformer with our STA module and apply this new **STA+Transformer** model for graph classification tasks. The experimental results are as follows:

|Long Range Graph Benchmark (LRGB) | | Peptides-func (graph classification)| Peptides-struct (graph regression) |
|--------------|------------:|------------:|-----------:|
|              | #Params. |  AP ↑  |  MAE ↓ |
| GCN|508k| 59.30±0.23| 34.96±0.13 |
| GCNII|505k| 55.43±0.78 | 34.71±0.10 |
| GINE |476k| 54.98±0.79| 35.47±0.45 |
| GatedGCN |509k| 58.64±0.77|34.20±0.13 |
| Transformer+LapPE |488k| 63.26±1.26| 25.29±0.16 |
| **STA**+Transformer+LapPE |488k|**65.83**±**0.94**| **24.16**±**0.21**|

By solely replacing the self-attention module with STA, we achieve noticeable improvements in performance. This experiment shows us the capabilities of STA in three different dimensions:
- STA can capture **long-range relationships** in these Long Range Graph Benchmarks. (Reviewer smZ8)
- STA can be applied to **graph-level** tasks and achieve competitive performance. (Reviewer ds4S)
- STA, as **a strong alternative to self-attention in the graph domain**, has the potential to replace and enhance existing attention-based graph learning methods.

---

We conclude by kindly inviting you to take a glance at this attached pdf. It intuitively illustrates how STA differs from vanilla graph attention methods.

---

### Decision · Program_Chairs · 2023-09-21

**Decision:**

Accept (poster)

**Comment:**

The authors design a novel Graph Transformer called STA-GNN (SubTree-attention-GNN) to address the over-smoothing and over-squashing in the message-passing scheme. The newly proposed Subtree Attention constructs for each node the similarity pairs of key and query matrices from its k-hop neighborhood. Experimental evaluations on node classification datasets confirm the effectiveness of STA and the competitive performance of STAGNN.
The reviewers agree that the paper is well-written and easy to follow. The idea of subtree attention is quite interesting and novel. The theoretical study shows that subtree attention can approximate global attention, which can be very useful for treating over-smoothing in MPNNs. Some concerns regarding the evaluation came up in the reviews but the additional experiments provided could address most of these concerns. After the discussions, the reviewers reached a consensus to accept this paper.